# SAFE-NID: Self-Attention with Normalizing-Flow Encodings for Network Intrusion Detection

**Brian Matejek**                                                                                           *brian.matejek@sri.com*
*Computer Science Laboratory, SRI International*

**Ashish Gehani**                                                                                           *ashish.gehani@sri.com*
*Computer Science Laboratory, SRI International*

**Nathaniel D. Bastian**                                                                              *nathaniel.bastian@westpoint.edu*
*Army Cyber Institute, United States Military Academy*

**Daniel J. Clouse**
*Laboratory for Advanced Cybersecurity Research, Department of Defense*

**Bradford Kline**
*Laboratory for Advanced Cybersecurity Research, Department of Defense*

**Susmit Jha**                                                                                             *susmit.jha@sri.com*
*Computer Science Laboratory, SRI International*

**Reviewed on OpenReview:** *https://openreview.net/forum?id=hDywd5AbIM*

## Abstract

Machine learning models are increasingly adopted to monitor network traffic and detect intrusions. In this work, we introduce SAFE-NID, a novel machine learning approach for real-time packet-level traffic monitoring and intrusion detection that includes a safeguard to detect zero day attacks as out-of-distribution inputs. Unlike traditional models, which falter against zero-day attacks and concept drift, SAFE-NID leverages a lightweight encoder-only transformer architecture combined with a novel normalizing flows-based safeguard. This safeguard not only quantifies uncertainty but also identifies out-of-distribution (OOD) inputs, enabling robust performance in dynamic threat landscapes. Our generative model learns class-conditional representations of the internal features of the deep neural network. We demonstrate the effectiveness of our approach by converting publicly available network flow-level intrusion datasets into packet-level ones. We release the labeled packet-level versions of these datasets with over 50 million packets each and describe the challenges in creating these datasets. We withhold from the training data certain attack categories to simulate zero-day attacks. Existing deep learning models, which achieve an accuracy of over 99% when detecting known attacks, only correctly classify 1% of the novel attacks. Our proposed transformer architecture with normalizing flows model safeguard achieves an area under the receiver operating characteristic curve of over 0.97 in detecting these novel inputs, outperforming existing combinations of neural architectures and model safeguards. The additional latency in processing each packet by the safeguard is a small fraction of the overall inference task. This dramatic improvement in detecting zero-day attacks and distribution shifts emphasizes SAFE-NID's novelty and utility as a reliable and efficient safety monitoring tool for real-world network intrusion detection.

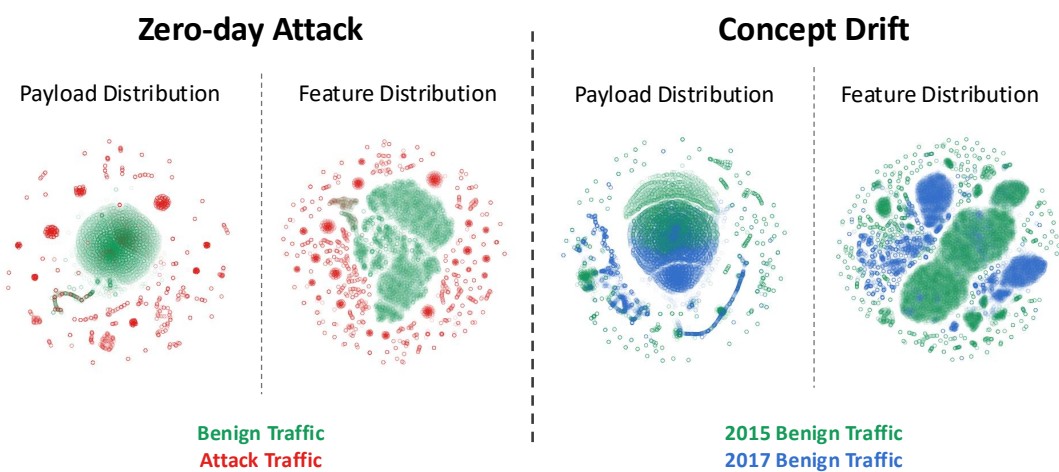

Figure 1: *Contrasting the input and feature space.* Deep neural networks trained on packet-level data can, with high accuracy, discriminate between benign and malicious traffic. However, these models fail in the presence of zero-day attacks and evolving network traffic. These drifts in payload byte distribution are often small, causing difficulty for OOD detection algorithms. We propose a "safeguard" generative model that learns the distribution of hidden layers from a DNN to identify novel inputs that the model cannot adequately evaluate. Shown above: t-SNE plots of the payload distributions and extracted feature distributions for benign and attack traffic (left) and evolving benign traffic (right).

# 1 Introduction

An increasing amount of new malware originates every day with both targeted and random attacks occurring against individuals, businesses, and government entities at large scales. These attacks often exploit known vulnerabilities or attempt to overwhelm system capacity to steal privileged information, disrupt legitimate traffic, and cause economic hardship to the victims. With an ever-increasing number of connected devices, manual monitoring of network traffic to identify malicious behavior is simply too expensive and infeasible. This has motivated increased adoption of machine learning (ML) (Leevy & Khoshgoftaar, 2020), in particular deep learning (DL) (Ferrag et al., 2020), to aid and increase the bandwidth of cybersecurity professionals.

Traditionally, ML models for network traffic operate at the *flow-level* (Wang, 2015; Kim et al., 2019; Sarhan et al., 2020). Network flow represents a continued connection between two devices parameterized by the five-tuple: (src_ip, src_port, dest_ip, dest_port, protocol). Flow summary tools such as CICFlowMeter (Draper-Gil et al., 2016; Lashkari et al., 2017) and Zeek (zeek) produce features from the bidirectional flows on which an ML model can be trained. These features include flow duration, bytes per second, and flag settings. These methods are inherently *ex post facto*, however, as the flow must conclude before we know the final statistics and decide whether it was benign or identify its attack category. Furthermore, the features themselves are highly variable to the flow capture mechanism and its setup (Sarhan et al., 2020). For these reasons, we focus on the *packet-level* classification pipeline that labels traffic into benign and different attack categories at a per-packet granularity. Expanding on existing work of packet-level classification (Shenfield et al., 2018; De Lucia et al., 2021; Bierbrauer et al., 2022; Bizzarri et al., 2024; Rani & Bastian, 2024), we include header context in inputs to our models to leverage additional information alongside the raw payload bytes. Furthermore, we design a light-weight encoder-only transformer model for the network intrusion detection task with a simple custom tokenization scheme.

While DNNs achieve high accuracy on inputs from the training distribution, they fail to generalize to novel inputs outside this distribution. Thus, their success on existing datasets does not entail their effectiveness in a real-world deployment where the network traffic is continuously evolving, and novel threats emerge routinely. We address this challenge by proposing a model-agnostic safeguard for DNNs that quantifies the uncertainty in the decision of any discriminative classification model by assigning a confidence score to each decision made by the DNN. The novel inputs are assigned low confidence and for such inputs, the

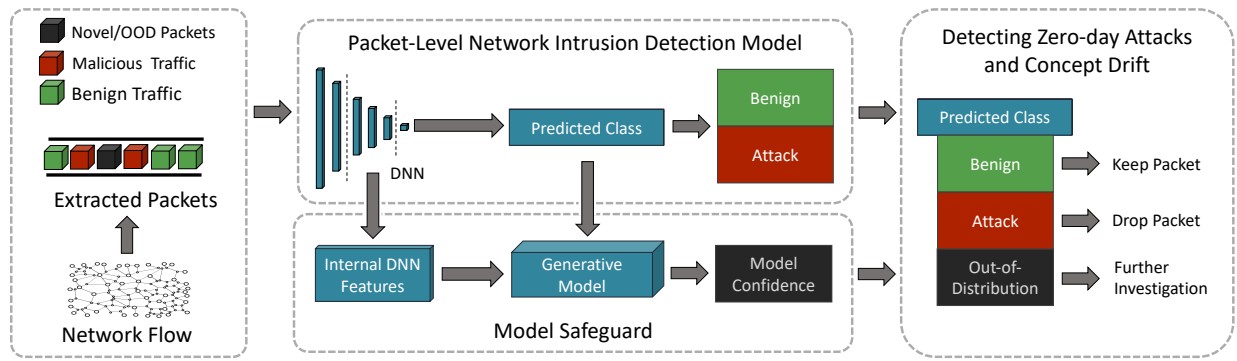

Figure 2: *Packet-level network intrusion detection system with model safeguards.* Our architecture ingests raw packet data and extracted information from the internet (IPv4) and transport layer (TCP/UDP) headers. We exclude information that is specific to the configuration used in data collection and not generalizable to the real-world. The predicted class and the internal features of the packet-level network intrusion detection model are used by the model safeguard. The monitor learns a class-condition distribution of the internal features and quantifies the uncertainty in prediction as a confidence score (that reflects the distance of internal features from the distribution of the features of the predicted class for any new input). The novel and out-of-distribution inputs are assigned low confidence scores and can, hence, be detected for further investigation. We show strong results on detecting novel inputs when using SAFE-NID—a transformer architecture for our discriminative classifier and normalizing flows for our model safeguard.

decision by the machine learning model cannot be trusted. A high rate of detection of novel inputs by the monitor ensures wrong decisions by the DNN are avoided, and a low false positive rate ensures that the needed bandwidth of intervention (such as human experts) remains low. Thus, the proposed model safeguard improves the robustness of the DNN network intrusion detection models. This approach differs from unsupervised anomaly detection algorithms which look for concept drift before inference (Rabanser et al., 2019; Han et al., 2023). We note that this is not an either-or situation. One could first detect anomalous inputs in an unsupervised manner before using a discriminative classifier. However, the classifier would still be susceptible to non-flagged out-of-distribution inputs, requiring a model safeguard. We learn the intermediate latent representations of the inputs as the in- and out-of-distribution traffic are more easily separable in the latent space (Figure 1). Furthermore, we highlight the success of SAFE-NID, which uses a combination of encoder-only transformer model for packet classification and normalizing flows for the model safeguard within this framework.

Without such robust models, system operators relying on DNN models may have overconfidence in their network security. For example, incorrectly labeled benign network traffic may contain malicious payloads. In particular, DNN models need to remain robust in the presence of novel inputs such as zero-day exploits and distribution shift as network traffic and payload distribution evolves. Although we cannot currently expect models to generalize to correctly classify all packets from zero-day exploits without additional finetuning, it is critical to recognize these novel inputs as out-of-distribution (Lee et al., 2018; Geng et al., 2020; Bulusu et al., 2020; Shen et al., 2021; Yang et al., 2021). By adding model safeguards to the DNN models, we can quantify uncertainty in the predictions from the models as a confidence score. Previous research has shown DNN models make overconfident wrong predictions on novel out-of-distribution inputs when considering softmax output (Guo et al., 2017) as the confidence score. Furthermore, any such uncertainty quantification should occur in near real-time (Cai & Koutsoukos, 2020) to augment current cybersecurity workflows where attacks can significantly impair operations within milliseconds (Shan et al., 2017).

In this paper, we present SAFE-NID, a novel framework designed to address these challenges. SAFE-NID employs a lightweight encoder-only transformer model, augmented with a safeguard that models uncertainty and ensures robust detection of both known and novel threats. By combining the efficiency of transformer models with the robustness of generative uncertainty quantification, our approach provides a significant advancement over existing methodologies. The overall uncertainty-quantified and robust deep learning ar-

chitecture proposed in this paper is described in Figure 2. We complement the DNN model used for detecting attacks with a model safeguard that learns the class-conditional distribution of the internal features of the DNN model over the training data. For any new input at inference time, the predicted class and the internal features from the DNN model are used to query the learned distribution model for the likelihood of the input being in-distribution. The confidence of the prediction is high for inputs which have higher likelihood. The proposed model safeguard architecture is compatible with any DNN model or architecture. We observe very high in-distribution accuracy for the network intrusion datasets for relatively simple DNN architectures such as feed-forward neural networks and convolution neural networks, as well as more complex architectures such as transformers. Our model safeguard and uncertainty quantification approach is agnostic to the DNN architecture, and could even be used with non deep learning approaches for latency restricted applications. However, we note best results with SAFE-NID: a light-weight encoder-only transformer architecture and normalizing flows safeguard within our framework.

We make the following contributions in this paper:

- We construct a network intrusion detection system that analyzes information at the packet-level. Traditional flow-level detection needs the flow to conclude before an attack can be detected. In contrast, the **packet-level detection** enables online nearly real-time detection of attacks and also avoids the need for engineering flow features (De Lucia et al., 2021; Bierbrauer et al., 2022).

- We train three DNN models, a CNN, FNN, and encoder-only transformer, on these datasets that achieve high accuracy. We empirically **demonstrate that these network intrusion deep learning models are not robust to inputs from novel attacks and concept drift**. The baseline models, despite having 99% accuracy on the in-distribution inputs, exhibit very poor accuracy $(< 1\%)$ **on novel attack classes**.

- We develop model **safeguards to protect our DNNs against OOD inputs and concept drift**. We show effectiveness in identifying suspect inputs using two different generative models (Lee et al., 2018; Papamakarios et al., 2021) to learn and represent the class-conditional distribution of internal features of the discriminative deep learning model being used for classification. The model safeguards can quantify uncertainty in the decision made by the underlying DNN models, and detect novel out-of-distribution inputs, despite the high imbalance of in-distribution to out-of-distribution samples $(600-2000 : 1)$.

- The light-weight encoder-only transformer discriminative classifier with our normalizing flows model safeguard in SAFE-NID outperforms existing baseline combinations of classifiers and out-of-distribution detection methods. Our **extensive empirical evaluation** over two network intrusion datasets (UNSW-NB15 and CIC-IDS-2017) considers different architectures (FNN, CNN, Transformer), and different OOD detection methods (Softmax, Energy-based, Gassian Kernel Density, and Normalizing Flow), and includes multiple ablation studies. The confidence score from our model safeguard achieves a high AU ROC of over 97% on detecting novel classes. This demonstrates how our framework is robust to new attack classes and novel inputs.

- We perform a detailed **latency analysis** across different DNN architectures and methods for detecting OOD inputs to demonstrate the practical feasibility of deploying the safeguards over DNN-based network intrusion detection systems. The latency of the safeguard is $(23 - 60 \, \mu \sec)$ which is a fraction of the inference latency of the DNN models $(144 - 2418 \, \mu \sec)$ demonstrating the low-overhead of the safeguard proposed in the paper and the potential of practical application of SAFE-NID for real-time network intrusion detection.

- We **create and release a packet-level network intrusion detection dataset** extracted from the flow-level datasets (Sharafaldin et al., 2018b; Moustafa & Slay, 2015), and make these available to the wider research community. We enumerate several challenges of using the flow-level datasets in this manner and provide guidance for extracting packet-level data from other similar datasets. We make our processed packet-level dataset freely available and our code open source to encourage further research on packet-level network intrusion detection.[1]

---

[1] https://github.com/SRI-CSL/trinity-packet

## 2 Related Works

A significant body of research concerns the identification of malware at different points of the attack pipeline. Early methods focused on identifying structural patterns within attributes from the PE headers in Windows executables (Saxe & Berlin, 2015; Anderson & Roth, 2018; Vinayakumar & Soman, 2018). More recent strategies attempt to exploit the advancements in computer vision by converting the raw binaries into images using N-grams (Raff et al., 2018; Mohammed et al., 2021). As an alternative, Ling *et al.* construct control graphs by unpacking and disassembling binaries (Ling et al., 2022). Although these methods achieve high classification accuracy, they are limited in scope as many frequent attacks exploit buffer overflows by tailoring inputs to otherwise benign programs (Butt et al., 2022). Furthermore, some disruption-oriented attacks such as the many variants of denial-of-service exploit the very processes that enable devices to interact like the SYN-ACK attack (Schuba et al., 1997). Network intrusion detection systems (IDS) classify network traffic as benign or malicious (Moustafa & Slay, 2015; Sharafaldin et al., 2018b). These setups can identify ongoing denial-of-service attacks as well as advanced persistent threats (APTs) that have already infiltrated a compromised system.

Most existing efforts on learning-based network intrusion focus on classifying traffic flows with several available datasets (Moustafa & Slay, 2015; Sharafaldin et al., 2018b). However, there are several difficulties in creating accurate training datasets for network intrusion detection such as adequately anonymizing data and correctly identifying malicious flows, among other problems. Therefore, the Canadian Institute for Cybersecurity (CIC) and Intelligent Security Group (ISG) at the University of New South Wales (UNSW) produced several datasets that capture both benign and malicious network traffic in small, simulated environments (Moustafa & Slay, 2015; Sharafaldin et al., 2018b; Koroniotis et al., 2019; Moustafa, 2021). These datasets typically provide raw pcap files summarizing all of the packets that pass through a victim network. Frequently, the authors provide bidirectional flow-level summaries produced by tools such as CICFlowMeter (Draper-Gil et al., 2016; Lashkari et al., 2017) or Zeek (zeek). These summaries include features such as flow duration, number of forward and backward packets, and number of flags set, among others.

A significant amount of research considers the problem of manually engineering features to capture traffic flows and using these features for the detection of malicious flows. Several different neural network architectures have been explored (Wankhede & Kshirsagar, 2018; Vinayakumar et al., 2019; Maxwell et al., 2019; Pelletier & Abualkibash, 2020) in addition to the traditional machine learning models such as AdaBoost (Yulianto et al., 2019) or random forests (Wankhede & Kshirsagar, 2018). Kim et al. (2019) train a CNN classifier on these features by transforming the 78-dimensional feature space into images. Since flow-level detection is *ex post facto*, real-time detection requires packet-level classification. There has been some work on packet-level detection of malicious traffic (Wang, 2015; Shenfield et al., 2018; De Lucia et al., 2021; Bierbrauer et al., 2022; Bizzarri et al., 2024; Rani & Bastian, 2024) wherein, the model takes as input the raw payload data, typically from the transport layer. Care must be taken to omit any potentially trivial information that could cause the ML model to learn the system configuration used in the data gathering process such as the IPs of the attack devices. This would lead to artificially high accuracy on the collected data which would not generalize to real-world environments. These previous approaches have generally focused on the discriminative models, with some analysis on transfer learning in the face of concept drift (Bierbrauer et al., 2022). However, they have not considered the problem of detecting inputs outside of the training distribution.

The problem of overconfident incorrect prediction by deep learning models on novel inputs that are out-of-distribution has been previously reported in domains such as computer vision (Guo et al., 2017) and several methods to compute confidence of deep learning models in these domains have been proposed (Hendrycks & Gimpel, 2016; Jiang et al., 2018; Jha et al., 2019; Ren et al., 2019). Broadly, these methods can be categorized into two classes. The first are the supervised techniques (Lee et al., 2017; Hendrycks et al., 2019; Meinke & Hein, 2019; Kaur et al., 2021) that require some exposure to the out-of-distribution inputs. This is impractical for cybersecurity where we cannot assume even a small number of packets corresponding to novel attack classes would be available at training time. Unsupervised approaches use only the in-distribution data. A direct approach is to use the softmax score of the classifier as a confidence metric (Hendrycks & Gimpel, 2016). ODIN (Liang et al., 2017b) enhances the softmax score by adding perturbations to the

input and using temperature scaling to the classifier's confidence. Recently, Macedo et al. (2021) proposed replacing softmax scores with isomax scores and entropy maximization for detection. Other unsupervised detection techniques based on energy scores (Liu et al., 2020b), trust scores (Jiang et al., 2018), and likelihood ratio (Ren et al., 2019) between the in-distribution and OOD data points have been proposed for detection. In contrast, we use a normalizing flow generative model for learning the distribution of internal features. These uncertainty-quantification and out-of-distribution detection approaches rely on learning the manifold or distribution of training data and are known to be susceptible to adversarial attacks (Jang et al., 2020).

One challenge with OOD detection using internal features (Hendrycks & Gimpel, 2017; Liang et al., 2017b; Liu et al., 2020b) is the phenomenon of feature collapse where a model's learned representations become overly similar, reducing the diversity of the feature space. This convergence can significantly impair out-of-distribution (OOD) detection methods. When feature collapse occurs, the feature space becomes less expressive, causing OOD samples to be mapped close to in-distribution (ID) features. This proximity results in high-confidence predictions for OOD inputs, thereby diminishing the effectiveness of OOD detection methods. There are several mitigation strategies to address this challenge. Contrastive learning methods (Wang & Isola, 2020) can be used to preserve feature diversity by pulling representations of similar samples closer while pushing dissimilar ones apart, ensuring the learned feature space remains expressive and suitable for distinguishing OOD samples. The feature collapse is more severe when there is a class imbalance, and this necessitates regularization to ensure minority classes remain fairly separable. One approach is to introduce a gradient penalty that enforces sensitivity to input changes, ensuring that the model remains responsive to input variations and maintains a diverse feature representation (Van Amersfoort et al., 2020). By adopting these techniques, models can mitigate feature collapse and maintain a more robust and expressive feature space, improving their ability to distinguish ID from OOD samples.

In this paper, we focus on robustness to novel inputs and out-of-distribution data. The goal is to build a detection pipeline where the generative safeguard assigns high uncertainty to recognize novel kinds of traffic or attacks which are different from the training data and on which the discriminative classifier cannot be trusted. Our attack model does not consider the scenario where an active attacker uses knowledge about our model safeguard and the detection pipeline to craft an attack specifically for our framework. The robustness to novelty is an important characteristic for building a robust ML-enabled system and is distinct from resilience to adversarial attacks. Our work is similar to novelty detection, out-of-distribution detection, and open-set recognition studied in vision and natural language domains. The robustness to novel and OOD is even more critical in cybersecurity applications, and our framework addresses this challenge.

## 3 Overview

We provide an overview of our system architecture in Figure 3. We take as input pre-existing flow-level IDS datasets that also provide raw pcap files (Section 4.1). Typically, these flow-level datasets use a flow summary tool to create a set of labeled features such as flow duration, and the number of forward and backward packets, among others. The flow labels come from existing knowledge of the test-bed architectures where set IPs and timestamps correspond to malicious activity of a certain type. Since the pcap files provided with the flow-level datasets are not labeled, we process the data to create packet-level datasets (Section 4.2). Since these existing datasets did not intend for this type of conversion, we identify a series of issues and solutions in this process. We convert the packets into input features for our neural networks by first taking the first 1,500 bytes of the payload, and second, extracting relevant features from the header (Section 4.3). We need to take care to not extract features that are trivially correlated with benign and malicious behavior, such as source or destination IP. We consider an encoder-only transformer neural network for the binary classification task for packets (Section 4.4). Although our transformer achieves high accuracies ($> 99\%$) on in-distribution data, we find that the networks perform poorly when we mimic novel inputs (e.g., zero-day exploits) (Section 4.5), or when the data consists of packets from a different year (Section 4.6). We extract internal features from the neural network and model their distributions using normalizing flows to produce a confidence (or novelty) score during inference (Section 4.7). This end-to-end system enables us to produce packet-level predictions with an uncertainty-quantification. Although we focus on SAFE-NID (the encoder-only transformer packet classifier and normalizing flows safeguard), the overall framework is agnostic to both models, allowing us to switch models based on real-world computational constraints.

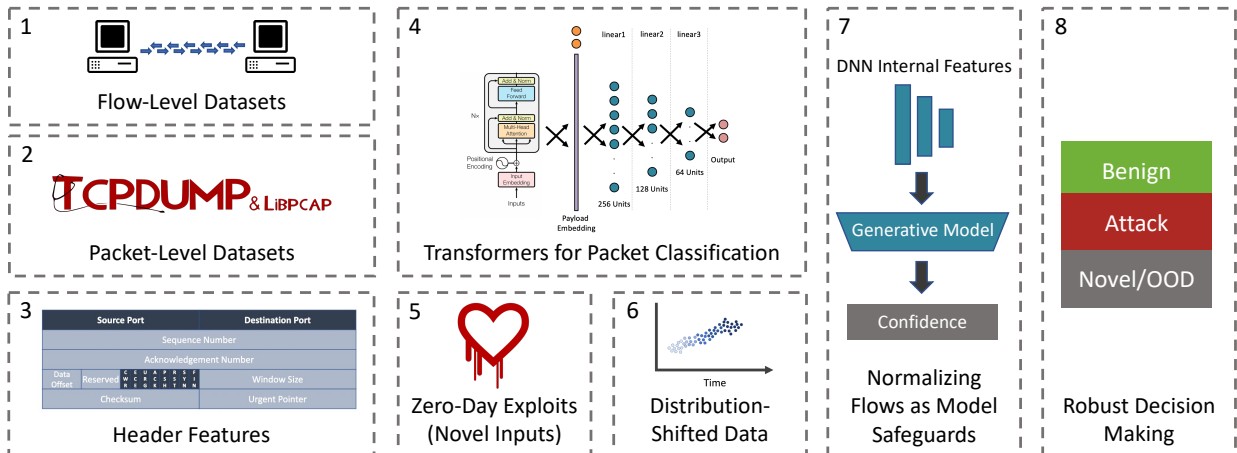

Figure 3: *System architecture overview.* We first take existing labeled flow-level datasets (box 1) and convert them into packet-level datasets by matching packets in a pcap file to their corresponding flows (box 2). We extract header features from the packets to augment our ML feature space, taking care to not include features that could trivially classify packets as benign or malicious, such as source and destination IP (box 3). A transformer take as input the payloads and header features to produce a binary classification of packets as benign or malicious (box 4). Despite high accuracy on the binary classification task, our model fails when given novel inputs, such as zero-day exploits (box 5) and inputs from a different time (box 6). We propose a normalizing flows model safeguard that takes the internal features from the DNN, models their distribution, and produces confidence in the predicted class (box 7). This enables us to have a robust decision-making process where we can classify inputs as benign, attack, or novel (box 8).

## 4 Technical Approach

Throughout our work, we use the following terminology: a *packet label* refers to the binary classification of a packet (i.e., benign or malicious), whereas a *packet category* refers to the multiclass classification of a packet that further granularlizes attack types (i.e., benign, denial-of-service, heartbleed, fuzzers, etc.).

### 4.1 Packet-level Capture

Most existing network IDS datasets provide flow summaries with accompanying benign and attack category labels (Moustafa & Slay, 2015; Sharafaldin et al., 2018b). We instead focus on classifying packets as benign and malicious for a couple of reasons.

First, flow capture devices extract different features from pcap files which creates difficulties when designing ML solutions that must work across a wide range of differently configured networks. Sarhan *et al.* use NetFlow (Claise, 2004) to extract features from four publicly available IDS datasets (Sarhan et al., 2020). NetFlow is easy to configure and generates features using almost solely packet headers; this ease of use, however, leads to a less expressive feature space compared to hand-tailored flow capture systems. In contrast, different packet capture technology running on different operating systems produce standardized PCAP files.

Second, learning benign and attack characteristics at the flow level is inherently a retroactive process. One has to wait for a flow to conclude before extracting features such as the number of forward and backward packets sent, mean payload length, and average time between packets, amongst others. Although this can be very useful in postmortem contexts where a network administrator wants to analyze fault points in a system's security after an attack concludes, it is a less viable paradigm for identifying ongoing and incoming attacks. By classifying packets as they arrive as benign or malicious, one can actively reject traffic before the payload arrives at its intended destination.

## 4.2 Preprocessing Data

We first must transform the raw pcap files into packet-level labeled datasets for training and inference. Since most of the existing literature on intrusion detection in network traffic focuses on flow-level features, existing datasets typically provide category labels (i.e., benign, denial-of-service, exploits, heartbleed, etc.) only at the flow-level.

Our method ingests the outputs from a flow capture system and the raw pcap files. We characterize each flow with the following five-tuple:

$$(\text{src\_ip, src\_port, dest\_ip, dest\_port, protocol})$$

Note that this five-tuple is often not discriminative enough on the IDS datasets since the victim and attack devices have set IPs by the architectural design of the testbed. That is, multiple flows will have the same five-tuple, and not all of those flows will have the same packet category or even label. Therefore, we need to not only identify a flow with the matching five-tuple but also select the flow with the proper timeframe.

Flow capture systems can be either unidirectional or bidirectional (Li et al., 2013). Bidirectional systems, such as CICFlowMeter (Lashkari et al., 2017; Draper-Gil et al., 2016) and NetFlow v9 (Claise, 2004), create a single flow once a host initiates a connection. These bidirectional flows cover any response packets to the initiating host. Therefore, these response packets have corresponding flows with the reciprocal of the standard five-tuple:

$$(\text{dest\_ip, dest\_port, src\_ip, src\_port, protocol})$$

Thus, we need to consider flows with both the standard five-tuple and its reciprocal during packet matching.

The pcap files record the packet timestamps in Coordinate Universal Time to microsecond precision. Note, we assume that the capture device has a correctly calibrated internal time. Unfortunately, flow summary tools do not have one standardized representation of the start and end times of the flows. Therefore, we once again need to be cognizant of the internal workings of the flow capture system. For example, the flows published with the CIC-IDS-2017 dataset use local start times (UTC-3) (Sharafaldin et al., 2018b) while those in the UNSW-NB15 dataset use universal time (Moustafa & Slay, 2015).

The disparity between the precision of the packet timestamps in the pcap files versus the precision of the start and end times in the flow files is perhaps as big an issue when accurately assigning packets to flows for category and label assignment. The flows that we consider for this work have time precision to the minute at worst and the second at best. However, the average duration of a flow is 1.48 ($\pm$3.37) and 0.66 ($\pm$13.93) seconds for the CIC-IDS-2017 and UNSW-NB15 datasets, respectively. The packet timestamps from the pcap files have microsecond precision. Therefore, we will often not identify any valid flows for a packet given the five-tuple with timeframe:

$$(\text{start\_time, start\_time} + \text{duration})$$

For example, we cannot match any packets that belong to a flow that starts at the half second (which the summary file rounds down) and only lasts a quarter second. Therefore, when matching packets to flows, we consider any five-tuple matching flow where the packet timestamp is in the range:

$$(\text{start\_time, start\_time} + \text{duration} + \text{precision})$$

where precision is the precision of the start\_time attribute of the flow summary files. However, this wider range of acceptable times often causes a given packet to match with several flows. We apply a conservative rule where a packet receives a packet category if and only if every valid flow has the same category as the one being applied to the packet. This prevents us from misclassifying packets to the wrong category or label.

We drop any packets that do not follow the IPv4 protocol at the internet layer and TCP or UDP protocols at the transport layer. Lastly, we drop all packets with empty payloads.

There are some other challenges and corner cases that we need to address to convert the CIC-IDS-2017 and UNSW-NB15 flow-level datasets into packet-level ones. We provide python code snippets for this process in

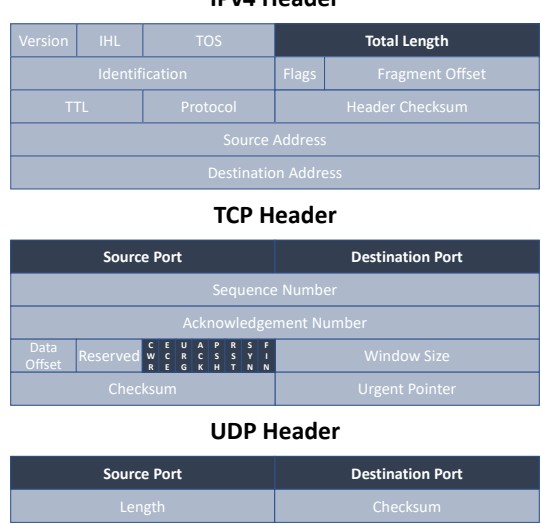

Figure 4: *Extracted header features.* We extract features from the packet headers (darkened fields) for our binary classification task. We do not include header information, such as source and destination IP, that could cause the system to learn the testbed architectures. We convert the source and destination ports into a binary vector that bins certain ports together (Table 1).

Appendix A. After preprocessing both datasets into a common format, we have a general purpose function that can take a series of processed flows and raw PCAP file and construct a packet-level dataset. We publish the code for this process and resultant data files[2].

## 4.3 Extracting Header Features

We extract information from the packet headers to augment our input feature space. However, we cannot simply input the entire header as our models will learn the setup configuration of the testbed architectures. For example, an ML model could trivially learn the source and destination IP addresses that correspond to devices in the attack network. Therefore, we carefully consider the header fields that provide useful contextual information that is transferable to other network configurations. We construct features from the darkened header fields shown for the IPv4, TCP, and UDP packet protocols in (Figure 4).

We only make use of the "Total Length" field in IPv4 headers. We divide this field by 65,536 to standardize the value between 0 and 1. Notably, we do not consider information in the "TTL", "Protocol", or "Address" fields. Although time-to-live might be useful in real-world applications with diverse network topologies, the publicly available datasets have very few plausible attack paths because of the limited number of victim and attack devices. We found that one of the most cited IDS datasets (Sharafaldin et al., 2018b) almost exclusively uses the TCP transport protocol for attacks (only two UDP attack packets). Therefore, we exclude the protocol field since the great imbalance is not indicative of the real world where some attacks, such as UDP Flood, use the UDP protocol (Dittrich, 1999). Lastly, most network IDS datasets have set IP addresses for the victim and attack devices. Training data with those fields would produce a model unable to generalize to other configuration settings.

For the transport layer headers, we produce a set of features based on the source and destination ports. We cannot simply produce a one-hot encoding for each port since many ports would indicate benign or attack based on the scripts that generate attacks on IDS datasets. After considering some commonly used network IDS datasets, we generate seven binary features for both the source and destination ports of every packet. These features are non-exclusive, i.e., at least one but perhaps two of the features can receive a value of one.

---

[2]https://github.com/SRI-CSL/trinity-packet

Table 1: *Port features for header context.* We construct seven indicator features from the source and destination ports. Features for sets that contain a given port number receive a value of one.

| Port Number | Description |
|:---:|:---:|
| 21 | File Transfer Protocol (FTP) |
| 22 | Secure Shell (SSH) |
| 80/8080 | Hypertext Transfer Protocol (HTTP) |
| 443/444 | Hypertext Transfer Protocol Secure (HTTPS) |
| 0 - 1,023 | Well Known Ports |
| 1,024 - 49,151 | Registered Ports |
| 49,152 - 65535 | Dynamic/Private Ports |

We can summarize the features as which of the sets in Table 1 include the port number. Note that we group ports 443 and 444—the heartbleed attacks on the CIC-IDS-2017 dataset exploit the SSL vulnerability on port 444. Lastly, we include the following eight flags from the TCP header: CWR, ECE, URG, ACK, PSH, RST, SYN, and FIN. We simply set these features to zero for UDP packets.

### 4.4 Encoder-Only Transformers for Packet Data

We consider an encoder-only transformer architecture for the sequence to classication task of predicting whether a payload is benign or malicious. We use a simple tokenization scheme to convert our payloads into vectors of tokens. We convert each byte in the payload into the corresponding ASCII number. Thus, we have a vocabulary size of 256 corresponding to the 256 character codes. We find existing subword tokenization strategies (Kudo & Richardson, 2018; Song et al., 2020) provide sub-optimal results on payload data, in part because encryption and compression create high entropy payloads that do not correlate to the structured text in natural languages. Our encoded payloads are then input into a series of transformer encoder blocks to create an embedding for each token (input byte) (Vaswani et al., 2017). We then use mean pooling to convert the array of embeddings into a single sentence embedding (Reimers & Gurevych, 2019). We concatenate the header context to the sentence embedding and input the resultant vector into three hidden fully connected layers with 256, 128, and 64 units. Each of the hidden layers has a LeakyRELU activation with $\alpha = 0.01$ (Maas et al., 2013) and batch normalization (Ioffe & Szegedy, 2015) and dropout regularization (Srivastava et al., 2014). The two output neurons have a softmax activation function.

### 4.5 Handling Zero-day Exploits

Machine learning models generally perform well when given in-distribution testing data, i.e., data similar to the training data. However, real-world cybersecurity defense implementations will eventually receive data that falls out of the training distribution. For example, zero-day exploits cannot by definition exist in the training data. It is imperative that any machine learning systems do not fail when such exploits appear. Here, we focus on false negatives where our model classifies packets from previously unseen attack paradigms as benign.

We see similar characteristics between the zero-day exploit problem and the open-set recognition one in the broader machine learning community (Geng et al., 2020). In open-set recognition, a model trains on incomplete data that does not include all classes that exist during inference. We model this behavior by training ML models on the benign traffic and all but one type of attack. We repeat this procedure for each attack category, to create $N$ trained models where each model corresponds to a single missing attack type. We consider two different metrics for success for our models: first, we can correctly classify the unseen attack types as malicious, and second, we can recognize that the attack types are unfamiliar and flag them as out-of-distribution. Correctly classifying unseen attacks as malicious indicates that our model is learning general attack patterns in the payloads themselves. For example, some of the brute force attack methods have similar characteristics and so removing one from training does not change the quality of inference. However, more often, when we remove an attack from the training data, we classify those packets on inference as benign. Thus, it is imperative to identify those packets as out-of-distribution (Section 4.7).

Table 2: *Dataset port distribution.* We extract seven binary features from the source and destination ports each. A feature has a value of one if the corresponding source/destination port belongs to the indicated set.

| Port(s) | UNSW-NB15 | | CIC-IDS-2017 | |
|---|---|---|---|---|
| **Source Port** | **No. Benign** | **No. Attack** | **No. Benign** | **No. Attack** |
| 21 | 689,008 | 19,288 | 40,417 | 17,975 |
| 22 | 1,592,854 | 284 | 128,195 | 43,651 |
| 80/8080 | 15,021,418 | 450,780 | 13,912,273 | 757,410 |
| 443/444 | 0 | 268 | 7,916,684 | 19,020 |
| Well Known (0 - 1023) | 19,186,346 | 1,015,630 | 24,025,879 | 837,249 |
| Registered (1024 - 49151) | 28,350,557 | 1,098,046 | 1,436,391 | 166,559 |
| Dynamic/Private (49152 - 65535) | 2,217,170 | 344,656 | 3,722,886 | 216,220 |
| **Destination Port** | **No. Benign** | **No. Attack** | **No. Benign** | **No. Attack** |
| 21 | 500,000 | 17,192 | 30,470 | 17,947 |
| 22 | 1,187,386 | 314 | 124,003 | 46,594 |
| 80/8080 | 354,910 | 112,911 | 589,520 | 286,401 |
| 443/444 | 0 | 394 | 2,305,603 | 31,026 |
| Well Known (0 - 1023) | 4,366,251 | 1,384,884 | 5,197,451 | 380,195 |
| Registered (1024 - 49151) | 34,715,404 | 789,883 | 3,618,349 | 440,957 |
| Dynamic/Private (49152 - 65535) | 10,672,418 | 283,565 | 20,369,356 | 398,876 |

## 4.6 Handling Distribution Shift

Internet most-common and best practices continually evolve, especially as privacy and security concerns become higher societal priorities. As the landscape morphs, the structure of the payloads changes, and previously trained models can become outdated. In recent years, an ever-increasing number of websites have transitioned from HTTP connections to HTTPS (signified by the rows for ports 80/8080 and 443/444 in Table 2). Sometimes this evolution occurs organically as more websites adopt existing libraries to improve privacy. In such instances, we expect a moderate distribution shift over small stretches of time. We model the natural distribution shift of network traffic by training on a dataset from Q1 2015 and inferring on one from Q3 2017, and vice versa. This 2.5-year difference corresponds to a significant period of change from unencrypted (HTTP) to encrypted web traffic (HTTPS), from approximately 30% in Q1 2015 to 60% in Q2 2017, per the percentage of web pages loaded by Firefox using HTTPS (letsencrypt). However, occasionally targeted government mandates or external business pressures encourage the rapid adoption of a new framework or protocol, such as when Google Chrome began labeling all HTTP connections as "Not Secured" in 2018 (Robbins, 2021). These "seismic" events can cause a significant break between the distributions of current network traffic from previous months (by Q2 2018, over 80% of web pages in the United States loaded by Firefox used HTTPS (letsencrypt)).

## 4.7 Normalizing Flows as a Model Safeguard

In the normalizing flows paradigm, a model learns a series of bijective transformations that can perform a one-to-one mapping from a simple distribution, such as a multivariate Gaussian, into a complex target distribution (Papamakarios et al., 2021; Kobyzev et al., 2020). Since each transformation block is invertible, we can take a feature vector and easily determine its corresponding value in the probability density function of the complex distribution by applying a series of matrix multiplications followed by the inverse of the simple activation functions. In this way, we extract features from the intermediate layers of our neural networks for the in-distribution training data and train normalizing flows that transform a multivariate Gaussian distribution into this complex space. During inference, we extract features from our network and transform the data into the multivariate Gaussian space. By looking at the loss (i.e., the probability that the vector belongs to the complex distribution), we can order our inputs by the probability that they belong to the training set distribution. Inputs with a lower negative log-likelihood loss are more likely to be in-distribution. We train two normalizing flows for each model, one each for benign and attack network

Table 3: *UNSW-NB15 packet distribution.* The UNSW-NB15 dataset contains over 52 million non-empty TCP and UDP payloads. The dataset captures nine attack categories as well as benign background traffic over two days.

| Packet Category | No. Packets | Avg. Payload Length |
|---|---|---|
| **Benign** | 49,754,073 | 2,140.15 ($\pm$1,195.68) |
| **Attack** | 2,458,332 | 2,236.90 ($\pm$1,140.02) |
| Analysis | 1,394 | 561.35 ($\pm$273.80) |
| Backdoor | 1,246 | 733.00 ($\pm$929.74) |
| DoS | 479,275 | 2,326.59 ($\pm$1,026.94) |
| Exploits | 1,619,721 | 2,250.03 ($\pm$1,148.98) |
| Fuzzers | 144,114 | 1,688.62 ($\pm$1,330.87) |
| Generic | 192,901 | 2,429.40 ($\pm$971.32) |
| Reconnaissance | 9,601 | 189.90 ($\pm$140.33) |
| Shellcode | 828 | 271.03 ($\pm$229.82) |
| Worms | 9,252 | 2,575.79 ($\pm$851.72) |

traffic. When evaluating the distance from our training distributions during inference, we only consider the normalizing flow model that matches the output label from our NN classifier.

There are several normalizing flow blocks common in the literature (Dinh et al., 2016; 2014; Kingma & Dhariwal, 2018; Sorrenson et al., 2020), with each offering advantages and trade-offs during training and forward and reverse inference. For our purposes, we use RealNVP blocks (Dinh et al., 2016) that we can parameterize with the following equation:

$$y = R\Psi(s_{\text{global}}) \odot \text{Coupling}\Big(R^{-1}x\Big) + t_{\text{global}} \tag{1}$$

where $R$ is a (deterministic) permutation matrix that allows each feature to influence the others, $\odot$ is the Hadamard Product (element-wise multiplication), $x$ is the input vector for each block, $s_{\text{global}}$ and $t_{\text{global}}$ are learnable parameters, and Coupling is the following function that first evenly divides the input vector into halves $x_1$ and $x_2$:

$$u = \text{concat}(u_1, u_2) \tag{2}$$
$$u_1 = x_1 \odot \exp\left(\alpha \tanh\left(s(x_2)\right)\right) + t(x_2) \tag{3}$$
$$u_2 = x_2 \tag{4}$$

$\alpha$ is a clamping value that restricts the range of possible values in the exponent, and $s$ and $t$ are learnable parameters. Note that in this coupling block, features in $x_2$ can influence outputs in $u_1$ but not vice versa. The permutation matrix allows each feature to influence the others when stacking multiple blocks during training.

To avoid overfitting to our training distribution, we add a small amount of Gaussian noise to our features during training and inference with the following equation:

$$X = X + \mathcal{N}(0, 0.05) \tag{5}$$

This improves the stability of the training procedure and simultaneously washes out any features with no discernible signal. Such features are not unexpected as others have previously observed feature collapse in deeper layers (Zhu et al., 2021).

## 5 Experimental Setup

### 5.1 Datasets

We evaluate our system using two different network intrusion detection datasets that publish both raw pcap data and corresponding hand-labeled flow data. Both datasets configure a testbed infrastructure with an

Table 4: *CIC-IDS-2017 packet distribution.* The CIC-IDS-2017 dataset contains over 30 million non-empty TCP and UDP payloads. The dataset captures fourteen attack categories as well as benign background traffic over the course of a week.

| Packet Category | No. Packets | Avg. Payload Length |
|---|---|---|
| **Benign** | 29,185,156 | $3,017.95\ (\pm 2,444.65)$ |
| **Attack** | 1,220,028 | $4,392.96\ (\pm 4,794.26)$ |
| Bot | 2,584 | $4,088.69\ (\pm 6,859.15)$ |
| DDoS | 220,160 | $6,166.16\ (\pm 5,822.03)$ |
| DoS GoldenEye | 26,198 | $5,484.42\ (\pm 5,509.26)$ |
| DoS Hulk | 747,340 | $4,810.35\ (\pm 4,467.81)$ |
| DoS Slowhttptest | 5,906 | $1,049.66\ (\pm 965.19)$ |
| DoS Slowloris | 23,593 | $404.80\ (\pm 204.47)$ |
| FTP-Patator | 35,922 | $44.06\ (\pm 14.41)$ |
| Heartbleed | 20,181 | $7,665.59\ (\pm 4,815.72)$ |
| Infiltration | 29,865 | $908.23\ (\pm 551.25)$ |
| Port Scan | 417 | $4,761.18\ (\pm 3,781.50)$ |
| SSH-Patator | 90,245 | $307.56\ (\pm 454.89)$ |
| Web Attack-Brute Force | 14,400 | $1,148.79\ (\pm 590.99)$ |
| Web Attack-SQL Injection | 19 | $2,511.68\ (\pm 1,966.98)$ |
| Web Attack-XSS | 3,198 | $2,280.02\ (\pm 1,398.51)$ |

attack and a victim network comprised of multiple connected devices. The attack networks initiate a series of different attack profiles throughout data collection.

**UNSW-NB15.** The UNSW-NB15 dataset (Moustafa & Slay, 2015) captures raw network traffic over two full (fifteen and sixteen hour) days in January and February 2015. In the testbed architecture, an IXIA traffic generator uses three virtual servers, two that spread benign network traffic and one that forms malicious activity. The servers pass all traffic through two routers connected to a firewall that allows all packets to pass through. The dataset authors installed tcpdump (Jacobson, 1989) on one of the routers to capture all packet data. The IXIA tool simulated nine different attack categories including analysis, backdoor, denial-of-service, exploits, fuzzers, generic, reconnaissance, shellcode, and worms. Table 3 provides an additional summary of each packet category after our processing method.

**CIC-IDS-2017.** The CIC-IDS-2017 dataset (Sharafaldin et al., 2018b) contains raw pcap data captured over a week-long period in July 2017. The devices on the victim network run different versions of the three most common operating systems (Windows, Mac, and Linux). Three of the attacking PCs have the Windows 8.1 operating system and the fourth has Kali Linux. The dataset authors configured one of the ports of the main switch as a mirror port that completely captures all traffic traversing into and out of the victim network. A B-Profile system profiles the abstract (benign) behavior of 25 different users and an automated agent derived from these profiles generates realistic benign events for packet capture (Sharafaldin et al., 2018a). The CIC-IDS-2017 dataset contains fourteen attack types from seven common attack families: brute force, heartbleed, botnet, denial-of-service, distributed denial-of-service, web, and infiltration. Table 4 provides a summary of the number of packets and average payload lengths for each packet category after our processing method.

We note that others (Engelen et al., 2021; Liu et al., 2022) have found issues with this particular dataset caused in part by bugs in the CICFlowMeter (Draper-Gil et al., 2016; Lashkari et al., 2017) tool. We avoid many of the troublesome artifacts by approaching the problem at a packet-level. For example, we drop any packets that have a "TCP appendix" error simply since we do not consider empty payloads. The other main artifacts come from attempted attacks that fail to deliver their malicious payloads. We still include these packets in our analysis, as watching an attack unfold remains a valuable goal, even before the malicious content arrives.

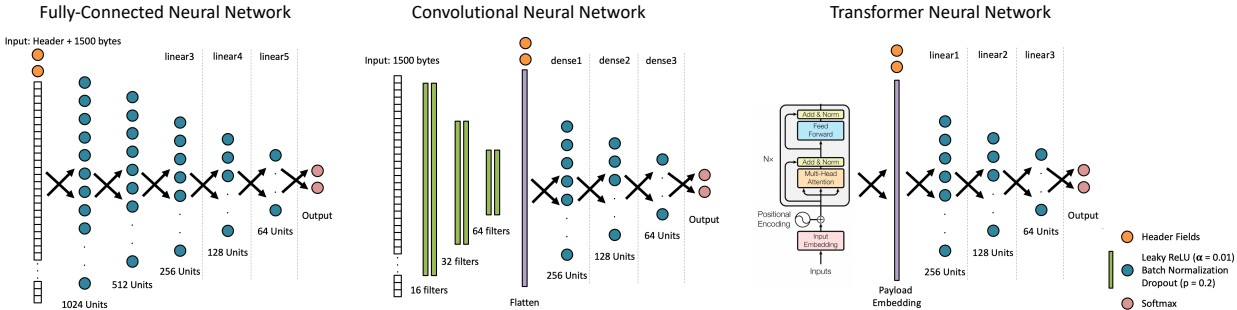

Figure 5: *DNN architectures for packet classification.* We train an FNN, CNN, and encoder-only transformer architecture for the packet label classification class. All networks take as input the first 1,500 bytes of the packet payload. We concatenate the header features to the payload for the FNN on input, the CNN after the last convolution, and the transformer after the sentence embedding. Schematic for transformer block is from Vaswani et al. (2017). All networks have two output neurons with a softmax activation.

## 5.2 Data Division

Both datasets have a high class imbalance with 20–25 times as many benign packets as attack ones. We split our data into training/validation and testing sets. For the training and validation data, we take half of the malicious samples and an equal number of benign ones. Of this data, we use 75% for training and 25% for validation. We take ten random splits in this fashion and show averages and standard deviations for all published results. Our test data contains the remaining malicious and benign samples. Thus, during inference, we have 39.5 and 46.8 times more benign packets than malicious ones for the UNSW-NB15 and CIC-IDS-2017 datasets, respectively. For the concept drift results, we use balance inference data with an equal number of benign and malicious packets for both datasets.

## 5.3 Baselines

### 5.3.1 Discriminative Classifiers

We consider fully-connected neural networks (FNNs) and convolutional neural networks (CNNs) as baselines for the binary classification task (De Lucia et al., 2021; Bierbrauer et al., 2022). For both architectures, we truncate payloads at 1500 bytes following practices in existing literature. Many ethernet networks have a maximum transmission unit size of 1500 bytes. We zero-pad payloads with fewer than 1500 bytes. We normalize all bytes to be between 0 and 1 for our FNNs and CNNs. Figure 5 summarizes the model architectures.

**Fully-connected Neural Network.** Our FNN model has five hidden layers starting with 1024 neurons and halving deeper into the network. We concatenate the header features to the 1500 payload bytes. Each neuron has a LeakyRELU activation with $\alpha = 0.01$ (Maas et al., 2013) and batch normalization (Ioffe & Szegedy, 2015) and dropout regularization (Srivastava et al., 2014). Our final output neurons use a softmax activation function.

**Convolutional Neural Network.** Our CNN model inputs the 1500 bytes into three blocks of two convolutions with a kernel size of three and 16, 32, and 64 filters, respectively, followed by a max-pooling by a factor of two. We flatten the output of the final max-pooling and concatenate the header features. We input the flattened vector into three hidden fully connected layers with 256, 128, and 64 units. Each of the hidden layers has a LeakyRELU activation with $\alpha = 0.01$ (Maas et al., 2013) and batch normalization (Ioffe & Szegedy, 2015) and dropout regularization (Srivastava et al., 2014). The two output neurons have a softmax activation function.

### 5.3.2 Model Safeguards

**Maximum softmax probability.** The maximum softmax probability (MSP) method for out-of-distribution detection uses the highest output from the final layer of the neural network (Hendrycks & Gimpel, 2017). This early baseline for out-of-distribution detection assumes a well calibrated model which will assign low probabilities for samples it cannot accurately classify. Since we have only two output classes, benign or attack, our normalized softmax probabilities range between 0.5 and 1.0.

**Energy-based detection.** We use as a baseline the energy-based out-of-distribution detection method from Liu et al. (2020a). We use our pre-trained neural networks (the three discriminative classifiers from the previous section). Note, even though we have a binary classification task on benign or malicious packets, we use two output neurons (one each for predicting whether a packet is benign or malicious). This allows differences in ranking of inference samples between the energy-based and MSP detection methods. Since we focus on zero-day exploits, we do not allow parameter tuning or fine tuning of our model safeguards.

**Gaussian kernel density.** Gaussian kernel density out-of-distribution methods obtain class conditional Gaussian distributions for the extracted intermediate features from our NNs (Lee et al., 2018). These methods assume that a class-conditional Gaussian distribution can adequately fit these features, and therefore inputs whose features are farther from the closest class-conditional Gaussian distribution are more likely to fall out-of-distribution (Lee et al., 2018). We approximate the covariance matrix of the dataset using the maximum likelihood estimate (empirical covariance). Once we obtain the covariance matrix, we use the Mahalanobis distance to measure the distance between our input points $P$ and the class-conditional distributions $D_i$ (Mahalanobis, 1936). Mahalanobis distances measure the number of standard deviations from $P$ to the mean of each $D_i$. We only calculate the distance for each point $P$ to the distribution corresponding to the classifier prediction (benign or attack).

### 5.4 Training Parameters

Our FNN and CNN baseline models each use batch normalization (Ioffe & Szegedy, 2015) and dropout ($p = 0.2$) (Srivastava et al., 2014) regularization techniques. Each hidden layer activation function is LeakyRELU ($\alpha = 0.01$). Both of the classification networks use the AMSGrad variant of the Adam optimizer (Kingma & Ba, 2014; Reddi et al., 2019) with $\beta_1 = 0.9$, $\beta_2 = 0.999$, and a learning rate of 1e−4. We use the binary cross-entropy loss function. We train each network for 20 epochs and use the weights with the lowest validation loss.

For our transformer architecture, we use an embedding size of 384, used previously in sentence transformer sequence to classification tasks (Reimers & Gurevych, 2019). We only stack two transformer blocks, and each block has six self-attention heads for 64-dimensional key, value, and query vectors (Vaswani et al., 2017). Similar to the FNN and CNN models, we use batch normalization (Ioffe & Szegedy, 2015) and dropout ($p = 0.2$) (Srivastava et al., 2014) regularization techniques for our fully connected layers after the transformer block. We use the AMSGrad variant of the Adam optimizer (Kingma & Ba, 2014; Reddi et al., 2019) with $\beta_1 = 0.9$, $\beta_2 = 0.999$, and a learning rate of 3e−4. We use the binary cross-entropy loss function and train each network for six epochs.

Since we focus on zero-day exploits, we, by problem construct, cannot fine tune our detection methods with known out-of-distribution samples. Thus, for our energy-based detection, we use a temperature of 1.0 to make it *parameter free* (Liu et al., 2020a).

For our normalizing flow models, we stack 20 RealNVP blocks with an affine clamping $\alpha = 2$. We learn our parameters for $s$ and $t$ using a simple fully connected network with two hidden layers with 128 features each and LeakyRELU activation with $\alpha = 0.01$. The input and output dimensions of these learnable blocks are dependent on the size of the extracted NN layers (either 256, 128, or 64 dimensions). We use the Adam optimizer (Kingma & Ba, 2014) with $\beta_1 = 0.8$, $\beta_2 = 0.9$, a learning rate of 1e−4, and weight decay of 2e−5. We train each normalizing flow model for 512 epochs. We use 25% of our in-distribution data as validation data, stratifying by packet category.

Table 5: *Discriminative classifier results.* Each neural architecture achieves a high accuracy on the binary packet-level classification task. The transformer architecture outperforms the others on the CIC-IDS-2017 dataset, whereas the FNN achieves the highest accuracy and F1-Score on the UNSW-NB15 dataset. These datasets are both highly imbalanced with 39.5 and 46.8× more benign samples than malicious ones.

| Architecture | Accuracy (↑) | AU ROC (↑) | F1-Score (↑) |
|---|---|---|---|
| **UNSW-NB15** | | | |
| CNN | 0.9950 (±0.0003) | 0.9997 (±0.0000) | 0.9952 (±0.0003) |
| FNN | **0.9951 (±0.0001)** | 0.9998 (±0.0000) | **0.9953 (±0.0001)** |
| Transformer | 0.9938 (±0.0011) | **0.9999 (±0.0000)** | 0.9941 (±0.0010) |
| **CIC-IDS-2017** | | | |
| CNN | 0.9940 (±0.0015) | 0.9955 (±0.0016) | 0.9944 (±0.0013) |
| FNN | 0.9926 (±0.0020) | 0.9955 (±0.0014) | 0.9931 (±0.0017) |
| Transformer | **0.9965 (±0.0011)** | **0.9973 (±0.0007)** | **0.9966 (±0.0010)** |

Table 6: *Modeling zero-day exploits.* The "In-Distribution" column refers to one model trained with all attack types and provides the recall for each attack on withheld testing data. The "Out-of-Distribution" column refers to 14 models with the indicated attack excluded from training. Some attack types have significant similarities with others and so excluding those does not greatly decrease their recall.

| Attack Category | No. Test Packets | In-Distribution (↑) | Out-of-Distribution (↑) |
|---|---|---|---|
| Bot | 1,292 | 0.9424 (±0.0270) | 0.0950 (±0.0935) |
| DDoS | 110,080 | 1.0000 (±0.0000) | 0.2145 (±0.0258) |
| DoS GoldenEye | 13,099 | 0.9999 (±0.0001) | 0.1995 (±0.0426) |
| DoS Hulk | 373,670 | 1.0000 (±0.0000) | 0.7529 (±0.1626) |
| DoS Slowhttptest | 2,953 | 0.9980 (±0.0025) | 0.5248 (±0.1659) |
| DoS Slowloris | 11,796 | 0.9996 (±0.0002) | 0.1737 (±0.2859) |
| **FTP-Patator** | **17,961** | **0.9994 (±0.0002)** | **0.0030 (±0.0019)** |
| Heartbleed | 10,090 | 0.4418 (±0.0660) | 0.0002 (±0.0002) |
| **Infiltration** | **14,932** | **0.9946 (±0.0051)** | **0.0780 (±0.0312)** |
| Port Scan | 208 | 0.9856 (±0.0071) | 0.8058 (±0.0234) |
| **SSH-Patator** | **45,123** | **0.9998 (±0.0001)** | **0.0027 (±0.0026)** |
| Web Attack-Brute Force | 7,200 | 0.9997 (±0.0001) | 0.9439 (±0.0701) |
| Web Attack-SQL Injection | 9 | 0.7222 (±0.1745) | 0.4678 (±0.1382) |
| Web Attack-XSS | 1,599 | 0.9959 (±0.0030) | 0.6305 (±0.1717) |

### 5.5 Implementation

We implement our system in Python using Payload-Byte (Farrukh et al., 2022) for extracting and labeling packet capture (PCAP) files of modern NIDS datasets, Pytorch (Paszke et al., 2019) for our neural networks and the Framework for Easily Invertible Architectures (FrEIA) (Ardizzone et al., 2018-2022) for our normalizing flows. For our energy-based and ODIN out-of-distribution detection methods we use the PyTorch Out-of-Distribution Detection Library (Kirchheim et al., 2022).[3] We publish our code to encourage further investigation by the research community.[4]

## 6 Results

### 6.1 Packet-level Classification Accuracy

Table 5 shows the accuracy for the all neural network architectures. In all configurations, our model accurately classifies packets with over 99.26% accuracy. These results come on highly imbalanced datasets, as discussed in Section 5.2. On the UNSW-NB15 dataset, the FNN outperform the CNN and transformer

---

[3]https://github.com/kkirchheim/pytorch-ood
[4]https://github.com/SRI-CSL/trinity-packet

Table 7: *Modeling distribution shift.* The overall accuracy on the UNSW-NB15 and CIC-IDS-2017 test sets decreases significantly when trained on the other. The left columns show the accuracy on the testing sets of each dataset when the corresponding training data is included during training. The right two columns show the inference accuracy when the training process excludes any examples from the given dataset.

| Architecture | Accuracy When In-Distribution | | Accuracy When Out-of-Distribution | |
|---|---|---|---|---|
| | **UNSW-NB15** | **CIC-IDS-2017** | **UNSW-NB15** | **CIC-IDS-2017** |
| CNN | 0.9969 (±0.0000) | 0.9923 (±0.0003) | 0.4861 (±0.0031) | 0.5221 (±0.0285) |
| FNN | 0.9968 (±0.0000) | 0.9913 (±0.0003) | 0.4733 (±0.0144) | 0.5114 (±0.0085) |
| Transformer | 0.9967 (±0.0006) | 0.9940 (±0.0010) | 0.4891 (±0.0095) | 0.4024 (±0.0363) |

architectures on the accuracy and F1-score metrics. The transformer architecture records the best results on all metrics on the CIC-IDS-2017 dataset.

## 6.2 Handling Zero-day Exploits

We reformulate the problem of classifying inputs from zero-day exploits into the more general open-set recognition task in machine learning. Using this strategy, we train models on all benign packets and all but one type of attack. We then can evaluate how well our model predicts packets from the withheld classes. In Table 6, the "In-Distribution" column shows the recall of predicting a packet from the category in the first column as an attack. For example, we classify 99.94% of FTP-Patator packets as malicious when we include FTP-Patator samples in the training data. The last "Out-of-Distribution" column shows the recall for a given attack when we exclude that attack from training while still including the other attacks.

We highlight three specific attack categories where recalls fall precipitously when excluded from training: FTP-Patator, Infiltration, and SSH-Patator. When included in the training, we classify between 99.46–99.98% of these malicious payloads. However, our models achieve accuracies between 0.27–7.80% when excluding the various classes from training. Perhaps more surprisingly, we can correctly classify several attack categories as malicious despite removing them from training, suggesting a general hierarchy of attacks where some are merely derivative of others. The WebAttack-BruteForce, DoS Hulk, and Port Scan attacks see only limited degradation in recall. We believe this occurs in part because these attacks develop in similar styles to others. Thus, we see limited degratdation in DoS Hulk, for example, because of similarities to either denial-of-service attacks.

We show the results for the CNN in Table 6. However, we note similar trends with the other configurations. We focus solely on FTP-Patator, Infiltration, and SSH-Patator when evaluating our real-time ML monitoring since those attack categories had both a significant number of packets and a large drop in recall when in- and out-of-distribution.

## 6.3 Handling Distribution Shift

We note substantial declines in accuracy when we train on one dataset and infer on the other (Table 7). In this table, the leftmost two results columns show the accuracies on the test set from the UNSW-NB15 and CIC-IDS-2017 datasets when we use the corresponding training data on our models. The rightmost two results columns show the accuracies when we train on one dataset and infer on the other (i.e., we train on UNSW-NB15 data and infer on the test set of the CIC-IDS-2017 dataset). Note, the accuracies do not match those from Table 5 since we use a balanced testing dataset with 50% benign and 50% attack packets. We do not find these results surprising since distribution shift is a well-known phenomenon in the vision domain (Taori et al., 2020; Kulinski & Inouye, 2022) with similar results in network traffic data (Bierbrauer et al., 2022). However, it does highlight the need for the safeguarding of any cybersecurity ML models.

Table 8 provides the breakdown for packet category accuracy. We note two opposite issues when inferring on the CIC-IDS-2017 and UNSW-NB15 datasets when trained on data from the other. Most of the CIC-IDS-2017 packets are labeled as malicious when inferred on models trained on UNSW-NB15 data. Thus, the vast majority of benign packets are erroneously labeled as attack. Conversely, most of the UNSW-NB15 packets

Table 8: *Out-of-distribution category-wise accuracy.* When inferring on the other dataset, we note significant decreases in accuracy. In particular, the models fail to classify correctly the benign packets in CIC-IDS-2017 and attack ones in UNSW-NB15.

| CIC-IDS-2017 | | UNSW-NB15 | |
|---|---|---|---|
| **Category** | **Accuracy (↑)** | **Category** | **Accuracy (↑)** |
| Benign | 0.1781 (±0.0112) | Benign | 0.9579 (±0.0278) |
| Bot | 0.7925 (±0.1436) | Analysis | 0.1056 (±0.1545) |
| DDoS | 0.7155 (±0.0309) | Backdoor | 0.0541 (±0.0488) |
| DoSGoldenEye | 0.6523 (±0.0469) | DoS | 0.0230 (±0.0244) |
| DoSHulk | 0.6604 (±0.1084) | Exploits | 0.0210 (±0.0160) |
| DoSSlowhttptest | 0.8478 (±0.0501) | Fuzzers | 0.0160 (±0.0099) |
| DoSSlowloris | 0.9224 (±0.0720) | Generic | 0.0087 (±0.0062) |
| FTP-Patator | 0.4878 (±0.0783) | Reconnaissance | 0.0363 (±0.0525) |
| Heartbleed | 0.7306 (±0.0204) | Shellcode | 0.0031 (±0.0068) |
| Infiltration | 0.8703 (±0.2108) | Worms | 0.0070 (±0.0070) |
| PortScan | 0.6378 (±0.1356) | | |
| SSH-Patator | 0.0005 (±0.0011) | | |
| WebAttack-BruteForce | 0.5268 (±0.0735) | | |
| WebAttack-SQLInjection | 0.4800 (±0.0588) | | |
| WebAttack-XSS | 0.5006 (±0.0100) | | |

are labeled as benign when inferred on models trained on CIC-IDS-2017. We again emphasize the drastic shift in most common protocols between the creation of the two datasets (Table 2). By 2017, most websites shifted to encrypted protocols such as HTTPS. Thus, very few of the benign examples in UNSW-NB15 are encrypted compared to the CIC-IDS-2017 dataset. This appears to have caused issues during inference for the CIC-IDS-2017 benign packets when out-of-distribution.

## 6.4 Detecting Out-of-Distribution Inputs

We evaluate our model safeguards with three metrics: area under the receiver operating characteristic (AU ROC), true positive rate at a true negative rate of 95% (TPR (TNR 95%)), and true positive rate at a true negative rate of 85% (TPR (TNR 85%)) (Tables 9 and 10). For the maximum softmax probability baseline, we order the inputs by the output value for the predicted class. Likewise, for the energy-based method, we take the output scores from the detector (i.e., the negative log sum of the exponential logit values, scaled by the temperature). We calculate the Mahalanobis distance and normalizing flow loss for each inference packet to assign a score. For Mahalanobis distance, normalizing flow loss, and energy-based score, higher values indicate more OOD-ness. For maximum softmax probability, a lower score indicates more OOD-ness.

For AU ROC, we order the inputs by increasing (or decreasing for MSP) score (farther from the target distribution) with binary labels of one corresponding to the out-of-distribution classes. For the TPR (TNR) scores, we adopt a hard threshold for what constitutes in- and out-of-distribution. Under these metrics, we allow ourselves to dispose of 5% and 15% of the in-distribution inference data (i.e., achieve a true negative rate of 95% and 85%). We then take the corresponding distance at those two thresholds and assign any inputs with scores greater than that as out-of-distribution. Our true positive rate thus refers to the proportion of out-of-distribution data that had a measured distance greater than those values. For all three metrics, higher values are better. We note that for some networks, losing nearly 5% of in-distribution data (including benign data), may not be tolerable. One can adapt these thresholds to the pain tolerances of the network users.

Table 9 shows the results for detecting out-of-distribution inputs using maximum softmax probability, energy-based, Gaussian kernel density, and normalizing flows. The final rows within each subproblem show the results using SAFE-NID. These tables contain the highest performing extracted layers for each network architecture. A complete set of results for all layers in the networks can be found in Appendix B. We note high AU ROC scores for Gaussian kernel density and normalizing flows with optimal values between 0.9668 and 0.9950 for each withheld attack class. Note, these results are with a high class imbalance with many

Table 9: *Out-of-distribution detection results for zero-day exploits.*

| Architecture | Layer | AU ROC (↑) | TPR (TNR 95%) (↑) | TPR (TNR 85%) (↑) |
|---|---|---|---|---|
| **FTP-Patator** | | | | |
| **Maximum Softmax Probability** | | | | |
| CNN | Output | 0.7931 (±0.1534) | 15.45% (±15.48%) | 64.19% (±37.97%) |
| FNN | Output | 0.6149 (±0.0893) | 0.12% (±0.37%) | 9.89% (±11.39%) |
| Transformer | Output | 0.4034 (±0.2232) | 0.21% (±0.54%) | 14.61% (±13.37%) |
| **Energy-Based** | | | | |
| CNN | Output | 0.7628 (±0.1657) | 12.51% (±12.78%) | 52.82% (±32.85%) |
| FNN | Output | 0.6060 (±0.1500) | 0.19% (±0.56%) | 16.55% (±19.86%) |
| Transformer | Output | 0.4133 (±0.1955) | 0.21% (±0.51%) | 15.98% (±17.58%) |
| **Gaussian Kernel Density** | | | | |
| CNN | dense3 | 0.8297 (±0.1157) | 26.47% (±24.29%) | 58.92% (±29.76%) |
| FNN | linear3 | 0.8473 (±0.0259) | 3.55% (±8.46%) | 59.20% (±19.59%) |
| Transformer | linear1 | **0.9950 (±0.0018)** | **100.00% (±0.00%)** | **100.00% (±0.00%)** |
| **Normalizing Flows** | | | | |
| CNN | dense2 | 0.8058 (±0.1224) | 4.53% (±5.42%) | 48.96% (±31.88%) |
| FNN | linear3 | 0.8640 (±0.0152) | 6.88% (±12.78%) | 61.51% (±14.50%) |
| Transformer | linear1 | 0.9936 (±0.0014) | **100.00% (±0.00%)** | **100.00% (±0.00%)** |
| **Infiltration** | | | | |
| **Maximum Softmax Probability** | | | | |
| CNN | Output | 0.6611 (±0.1897) | 26.16% (±20.54%) | 53.95% (±23.83%) |
| FNN | Output | 0.7676 (±0.0567) | 12.39% (±11.58%) | 47.52% (±16.07%) |
| Transformer | Output | 0.3167 (±0.2055) | 3.37% (±6.66%) | 9.56% (±19.88%) |
| **Energy-Based** | | | | |
| CNN | Output | 0.6744 (±0.1859) | 25.33% (±21.43%) | 53.56% (±26.27%) |
| FNN | Output | 0.7592 (±0.0750) | 11.41% (±10.28%) | 47.91% (±15.66%) |
| Transformer | Output | 0.3037 (±0.2291) | 4.33% (±11.43%) | 10.13% (±23.19%) |
| **Gaussian Kernel Density** | | | | |
| CNN | dense3 | 0.9622 (±0.0217) | **79.11% (±15.19%)** | 96.29% (±2.99%) |
| FNN | linear4 | 0.8748 (±0.0405) | 41.70% (±15.30%) | 61.45% (±16.68%) |
| Transformer | linear2 | 0.9552 (±0.0318) | 71.26% (±28.99%) | 94.93% (±13.73%) |
| **Normalizing Flows** | | | | |
| CNN | dense3 | 0.9638 (±0.0153) | 77.08% (±17.79%) | 98.22% (±0.78%) |
| FNN | linear4 | 0.8820 (±0.0300) | 30.41% (±15.53%) | 67.72% (±14.22%) |
| Transformer | linear2 | **0.9668 (±0.0182)** | 77.89% (±23.41%) | **99.18% (±2.25%)** |
| **SSH-Patator** | | | | |
| **Maximum Softmax Probability** | | | | |
| CNN | Output | 0.7821 (±0.0511) | 17.19% (±10.35%) | 66.61% (±15.90%) |
| FNN | Output | 0.6568 (±0.0727) | 1.53% (±0.87%) | 31.75% (±11.64%) |
| Transformer | Output | 0.1924 (±0.1103) | 0.20% (±0.35%) | 2.24% (±2.45%) |
| **Energy-Based** | | | | |
| CNN | Output | 0.7807 (±0.0701) | 18.90% (±9.81%) | 64.77% (±14.63%) |
| FNN | Output | 0.5932 (±0.0896) | 1.38% (±0.92%) | 23.38% (±12.69%) |
| Transformer | Output | 0.1833 (±0.1142) | 0.19% (±0.34%) | 2.12% (±2.57%) |
| **Gaussian Kernel Density** | | | | |
| CNN | dense3 | 0.7254 (±0.1269) | 23.72% (±10.20%) | 46.08% (±12.92%) |
| FNN | linear3 | 0.8967 (±0.0100) | 26.14% (±4.38%) | 77.99% (±5.01%) |
| Transformer | linear1 | 0.9821 (±0.0037) | 94.90% (±2.61%) | 99.05% (±1.09%) |
| **Normalizing Flows** | | | | |
| CNN | dense2 | 0.7086 (±0.1117) | 11.23% (±4.00%) | 39.33% (±13.11%) |
| FNN | linear3 | 0.9119 (±0.0062) | 30.81% (±3.04%) | 84.98% (±3.99%) |
| Transformer | linear1 | **0.9901 (±0.0007)** | **100.00% (±0.00%)** | **100.00% (±0.00%)** |

Table 10: *Out-of-distribution detection results for concept drift.* We achieve slightly lower AU ROC scores on the out-of-distribution CIC-IDS-2017 and UNSW-NB15 datasets, in part because we have higher overall accuracy (more overlap of distributions). Additional results can be found in Appendix B.

| Architecture | Layer | AU ROC (↑) | TPR (TNR 95%) (↑) | TPR (TNR 85%) (↑) |
|---|---|---|---|---|
| **CIC-IDS-2017** | | | | |
| **Maximum Softmax Probability** | | | | |
| CNN | Output | 0.7818 (±0.0254) | 36.43% (±4.48%) | 63.02% (±4.47%) |
| FNN | Output | 0.7187 (±0.0289) | 23.25% (±3.21%) | 50.69% (±3.18%) |
| Transformer | Output | 0.6731 (±0.0737) | 23.41% (±8.93%) | 38.37% (±14.72%) |
| **Energy-Based** | | | | |
| CNN | Output | 0.7589 (±0.0436) | 36.79% (±6.40%) | 60.53% (±3.58%) |
| FNN | Output | 0.7114 (±0.0498) | 22.90% (±3.63%) | 49.71% (±3.14%) |
| Transformer | Output | 0.6419 (±0.0966) | 21.88% (±9.79%) | 35.48% (±14.45%) |
| **Gaussian Kernel Density** | | | | |
| CNN | dense2 | 0.8981 (±0.0170) | 55.39% (±5.94%) | 77.16% (±5.28%) |
| FNN | linear4 | 0.7343 (±0.0597) | 34.96% (±4.17%) | 45.25% (±5.73%) |
| Transformer | linear2 | 0.8753 (±0.0552) | 58.05% (±11.80%) | 73.07% (±11.11%) |
| **Normalizing Flows** | | | | |
| CNN | dense2 | 0.8960 (±0.0151) | 52.88% (±3.63%) | 76.19% (±5.61%) |
| FNN | linear3 | 0.8130 (±0.0226) | 27.27% (±6.58%) | 50.10% (±2.03%) |
| Transformer | linear2 | **0.9259 (±0.0046)** | **73.01% (±2.10%)** | **81.55% (±1.11%)** |
| **UNSW-NB15** | | | | |
| **Maximum Softmax Probability** | | | | |
| CNN | Output | 0.5590 (±0.0063) | 34.87% (±2.45%) | 44.47% (±3.19%) |
| FNN | Output | 0.7349 (±0.0417) | 25.98% (±2.04%) | 43.43% (±5.41%) |
| Transformer | Output | 0.6255 (±0.1033) | 24.21% (±11.24%) | 40.14% (±14.78%) |
| **Energy-Based** | | | | |
| CNN | Output | 0.5381 (±0.0480) | 32.24% (±5.82%) | 41.78% (±8.91%) |
| FNN | Output | 0.7177 (±0.0977) | 26.62% (±2.73%) | 44.61% (±8.65%) |
| Transformer | Output | 0.6153 (±0.1003) | 24.26% (±11.29%) | 39.55% (±14.19%) |
| **Gaussian Kernel Density** | | | | |
| CNN | dense2 | 0.9263 (±0.0139) | 61.61% (±4.44%) | 83.14% (±4.44%) |
| FNN | linear4 | 0.9079 (±0.0109) | 56.06% (±5.32%) | 80.31% (±2.68%) |
| Transformer | linear2 | **0.9636 (±0.0126)** | **79.23% (±8.25%)** | **95.73% (±2.68%)** |
| **Normalizing Flows** | | | | |
| CNN | dense2 | 0.9263 (±0.0063) | 55.98% (±1.29%) | 83.03% (±2.72%) |
| FNN | linear4 | 0.8963 (±0.0112) | 45.10% (±8.46%) | 77.66% (±3.28%) |
| Transformer | linear3 | 0.9583 (±0.0090) | 77.71% (±4.49%) | 94.44% (±2.24%) |

more benign samples than attack ones. Therefore, our imbalance of in-distribution to out-of-distribution samples for FTP-Patator, Infilration, and SSH-Patator is 1623.9 : 1, 1953.5 : 1, and 645.8 : 1, respectively.

Most of the FTP-Patator and SSH-Patator payloads are simply zeros causing a large amount of redundancy in the feature space before the dense layers for the CNN (with average payload lengths of 44.06 (±14.41), 307.56 (±454.89), respectively). Perhaps unsurprisingly then, we note that it is better to extract features from the FNN for FTP-Patator and SSH-Patator than the CNN, although best yet to use the transformer architecture. We note the opposite trend for the CNN and FNN for Infiltration, which has an average payload length of 908.23 (±551.25). The transformer architecture with both Gaussian kernel estimation and normalizing flows performs well for all OOD inputs. The maximum softmax probability and energy-based methods performs poorly across the board with no high scores for any attack class.

Table 10 shows our results for detecting the CIC-IDS-2017 and UNSW-NB15 datasets as OOD. The last rows for each dataset grouping contain results for SAFE-NID. We note lower AU ROC scores than for the withheld CIC attack classes. We attribute this, in part, to the higher accuracies when these datasets are left out of training. Although there was a distribution shift between the datasets, we still achieve higher accuracies. Thus some of these packets may not truly be out-of-distribution. The transformer architecture outperforms the others during this task as well, with the normalizing flows and Gaussian kernel density safeguards performing best on CIC-IDS-2017 and UNSW-NB15, respectively.

Across all scenarios, the OOD methods that model the internal features of the neural networks outperform the output-based methods. Thus, we assert extracting the features and running the model safeguard warrants the increased computational costs associated with both methods. Furthermore, both methods do not require outlier exposure, which is critical in the case of zero day exploits. We found poor results with ODIN (Liang et al., 2017a) without the step of parameter tuning and network finetuning with outlier exposure (Appendix C).

Appendix B provides the complete list of metrics for each OOD class (e.g., FTP-Patator), OOD method, neural architecture, and extracted layer (for Gaussian kernel density and normalizing flows). Considering the AU ROC values for zero day exploits, we generally note some feature collapse for the FNN and Transformer architectures, affecting the scores when using the last hidden layer. In contrast, we find higher scores when using the last two hidden layers than the preceding layer for the CNN architecture (with the exception of Infiltration paired with Gaussian kernel density safeguard where all three scores are similar). Thus, for future datasets, we would suggest focusing on the intermediate hidden layers to avoid the degradation is safeguard accuracy. In our concept drift use cases, we see less clear evidence of feature collapse affecting the results of our safeguard with some combinations of neural architecture and safeguard performing best in the later layers.

Generally, normalizing flows outperforms Gaussian kernel density when modeling the internal features from the FNN and Transformer classifiers. For the CNN, we find better results when using the Gaussian kernel density modeling for the internal features. We note, however, that both methods do generally perform similarly well and the differences in result are often quite small between the two safeguards. Uniformly across all experiments we find that using a transformer for the binary classification task followed by either normalizing flows or Gaussian kernel density safeguards. Both methods significantly outperform the baseline methods that use only the output logits. Thus, modeling the internal features of the discriminative classifier is an important step in any defensive system.

## 6.5   Normalizing Flows Ablation Study

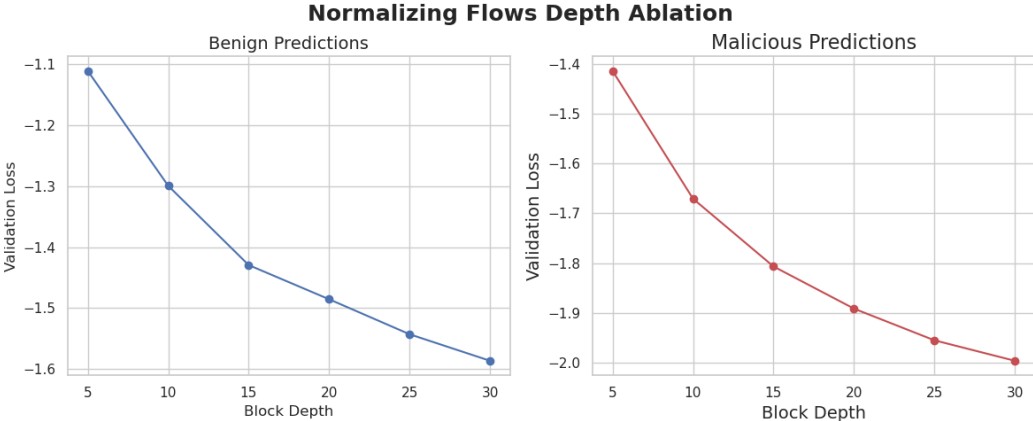

Figure 6: The validation loss decreases with the number of blocks in the normalizing flows model safeguard. However, the cost of this safeguard increases with the number of blocks. Thus, we choose 20 blocks as an acceptable tradeoff of performance and computation.

Table 11: *Model safeguard latency.* Average number of microseconds to process a packet for the model safeguards

| Gaussian Kernel Density | | Normalizing Flows | |
|---|---|---|---|
| Feature Dimensionality | Latency ($\downarrow$) | Feature Dimensionality | Latency ($\downarrow$) |
| 256 | 59.60 µ sec | 256 | 40.45 µ sec |
| 128 | 29.59 µ sec | 128 | 35.37 µ sec |
| 64 | 23.96 µ sec | 64 | 32.95 µ sec |

We perform an ablation study on some of the hyperparameters from our normalizing flows model safeguard (namely, number of blocks and clamping value). For our ablation study, we focus on the Infiltration zero-day exploit. Compared to FTP-Patator and SSH-Patator, all three neural architecture paired with normalizing flows had room for improvement. We find that varying clamping values between $[0.5, 3.0]$, in increments of 0.5, has nearly no effect on the validation loss. For this experiment, we stratified by neural architecture, layer extracted, number of blocks, and classifier output layer, while varying the clamping value. We find that on average a clamping value of 2.0 yields a validation loss that is within 1.24% of the optimal clamping value for that suite of experiments. In over 90% of these experiments, a clamping value of 2.0 was within 3% of the optimal clamping value found.

We perform an additional study on the effect of changing the number of blocks in our normalizing flows safeguard. We consider between 5 and 30 blocks, in increments of 5 (Figure 6). The validation loss decreases as the number of blocks increases, with big improvements between 5 and 20 blocks. The level of increase tapers as the number of blocks increases. This comes at inference time expense. Thus, we use 20 blocks as an acceptable tradeoff of results and computational cost.

## 6.6 Latency Analysis

The CNN, FNN, and transformer architectures require 172.58 µ sec, 144.60 µ sec, and 2418.02 µ sec to process one packet, respectively. We note that the transformer architecture takes significantly more time during inference than the other two networks, with the fully-connected neural network requiring the least overhead.

We show the amount of processing time per packet in Table 11 for our model safeguards. The feature dimensionality depends on the extracted layer from our DNNs. Both safeguard implementations take on the order of tens of microseconds to process each packet. The timing analysis was run on an AMD Ryzen Threadripper PRO 5965WX 24-Cores processor at 1.8 GHz for Gaussian kernel density and on an NVIDIA RTX A6000 for normalizing flows. The average time from packet to flow termination in CIC-IDS-2017 is 31.29 seconds (median time is 11.56 seconds). Thus, packet-level detection has very low latency when compared with flow-level detection. The maximum softmax probability and energy-based methods require limited additional overhead cost as it merely takes the output from the last layer. The ODIN method requires two forward passes and one backwards one causing significantly higher computational overhead, particularly for the transformer architecture.

We note a trade off between latency and accuracy. A real-world implementation of our system might employ different classifiers on network traffic depending on the perceived importance of the receiving devices. For example, an email server or data repository with sensitive personally identifiable information might use SAFE-NID as its defensive apparatus. Some less sensitive use cases that demand higher throughput (e.g., the population of products from a query on an e-commerce website) might use the FNN with an MSP OOD detector. We have not focused on current methods for reducing the computational costs during inference for neural networks such as quantization (Jacob et al., 2018; Choukroun et al., 2019), pruning (Han et al., 2015), or model distillation (Hinton, 2015). However, we can foresee a future where one uses these optimization strategies for both the discriminative classifier and model safeguard. One would similarly identify which strategies provide a sufficient increase in throughput with limited degradation in accuracy.

## 6.7 Broader Impact Statement

We note some practical and privacy concerns when identifying malicious activity at the packet level as opposed to at the flow level. A real-world implementation of this system would need to protect user data privacy by encrypting sensitive payloads, minimizing metadata, and complying with privacy laws such as the General Data Protection Regulation (GDPR) and the California Consumer Privacy Act (CCPA). More generally, we do not foresee the deployment of such a system outside of enterprise networks where security is paramount (and privacy is a secondary concern). Still, such a system could operate at the end points of communication to reduce any privacy concerns from employees. This does introduce some practical issues such as latency (Section 6.6) introduced by both the discriminative classifier and model safeguard. Thus, depending on the importance of the data, a functioning system could decide which classifier/safeguard combination to use by prioritizing accuracy or throughput. The emergence of low-cost and high-throughput specialized hardware for ML inference encourages the practical adoption of ML-based network intrusion detection, and several such methods have already been deployed, such as Cisco Secure Network Analysis[5], CrowdStrike ExtraHop[6], Darktrace[7], and Sangfor Cyber Command[8]. Our approach provides a lightweight safeguard to detect OOD inputs to such ML models.

## 7 Conclusions

There are several challenges to using deep learning methods for classifying packets as benign or malicious, including the creation of such datasets from existing publicly available flow-level datasets. We identify these challenges and publish the data for future research. We demonstrate that DNNs trained for network intrusion detection on these datasets achieve high accuracy (over 99%) on in-distribution inputs, but fail drastically (accuracy below 1%) in the presence of zero-day attacks, with severe degradation in the presence of concept drift. We address this robustness challenge by constructing a model safeguard that uses the prediction of the DNN classifier and its internal features to quantify the uncertainty in predictions made by the classifier and to detect novel and out-of-distribution inputs. We propose SAFE-NID: a lightweight encoder-only transformer architecture for the packet classification task with a normalizing flows model safeguard. The normalizing flows model learns the distribution of internal features of the neural networks on in-distribution input samples. We then compare features from new inputs to this distribution to identify out-of-distribution examples. Our proposed solution achieves an AU ROC score of over 0.9668 in detecting novel inputs from withheld attack classes.

While SAFE-NID presents a robust approach to packet-level intrusion detection, some limitations highlight opportunities for further enhancement and future research. A notable limitation lies in the computational complexity introduced by employing transformer models and normalizing flows, particularly in high-throughput network environments where latency is a critical factor. Table 11 shows that our approach takes microseconds to process a packet and we identify further reduction of this latency as a promising direction of future research. Moreover, while our analysis demonstrates robust evaluations under both in-distribution and out-of-distribution settings, the dynamic nature of modern network environments presents additional challenges, such as handling real-time packet overlaps, session mismatches, and diverse attack surfaces. Future research could focus on adapting SAFE-NID to dynamically evolving network conditions and further improving its resilience to such variations. Another promising area for future exploration involves enhancing the system's robustness against adversarial attacks. While SAFE-NID demonstrates efficacy in detecting zero-day attacks, adversarially crafted packets designed to evade detection remain a significant threat. Incorporating adversarial robustness into the framework could help address this gap. Additionally, SAFE-NID's reliance on packet-level analysis, while advantageous for low-latency detection, might miss contextual insights available at the flow or session level. Combining packet-level and flow-level approaches, or developing hybrid models, could provide a more holistic perspective and improve detection performance. Addressing these limitations not only enhances the current framework but also opens avenues for advancing the state-of-the-art in intrusion detection systems tailored to modern, fast-evolving network environments.

---

[5]https://www.cisco.com/c/en/us/products/collateral/security/stealthwatch/datasheet-c78-739398.html
[6]https://assets.crowdstrike.com/is/content/crowdstrikeinc/crowdstrike-extrahop-store-partner-data-sheetpdf
[7]https://darktrace.com/resources/darktrace-network-product-brief
[8]https://www.sangfor.com/cybersecurity/products/cyber-command-ndr-network-detection-and-response

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

# A   Preprocessing Data

We provide Python code for converting a flow-level dataset into a packet-level one. We also provide code to convert both the CID-IDS-2017 and UNSW-NB15 flows into a unified data structure for processing into a general purpose data transformer. Our code is also freely available.[9]

## A.1   Converting Flow-Level Data in Dictionaries

Flow-level datasets are typically comprised of one or more spreadsheets with columns associated with IP addresses, port numbers, protocols, timestamps, durations, and labels, among other things. This function produces a dictionary of these flows with which to pair PCAP packets. The following function parses these datasets, extracts all relevant flow information, and produces dictionaries where the keys are of the form:

$$(\text{src\_ip, src\_port, dest\_ip, dest\_port, protocol})$$
$$(\text{dest\_ip, dest\_port, src\_ip, src\_port, protocol})$$

and the values corresponding to a list of flows (with timestamps, category label, and timestamp precision) that match this five-tuple. Since the flows are bidirectional in the spreadsheets, we need to record all flows with both the source and destination IP addresses and ports first. Otherwise we would miss any response packets, since the spreadsheets only acknowledge the initiator of the connection as the source, whereas in packet data, the source and destination fields oscillate based on the sender and receiver. There are a few miscellaneous artifacts of the datasets that we otherwise need to correct for before saving our dictionary. We sort the arrays for each dictionary value by the end time of the flow to produce a ordering of flows that finish first. The next two subsections provide additional information on the conversion of a spreadsheet into entries into the dictionary for both the UNSW-NB15 and CIC-IDS-2017 datasets, respectively.

```python
def generate_flow_labels_hash(dataset):
    """
    Generate a hash for flows to labels with the given tuple:
    (source_ip, destination_ip, source_port, destination_port).

    @param dataset: ContextPCAPDataset to generate hashes for
    """
    # create a dictionary of flow hashes and a counter of the number of flows seen
    flow_labels = {}
    nflows = 0

    # read all of the flow CSV files
    for flow_filename in dataset.flow_datasets:
        flow_start_time = time.time()
        sys.stdout.write('Parsing flow file {}...'.format(flow_filename))

        if dataset.flow_preprocessor == 'CIC-IDS':
            data = pd.read_csv(flow_filename, delimiter = ',', encoding = 'cp1252')

            nflows += cic_ids_flow_processor(dataset, data, flow_labels, nflows)
        elif dataset.flow_preprocessor == 'UNSW-NB15':
            data = pd.read_csv(flow_filename, dtype = {'attack_cat': str})

            # attack category left blank for normal traffic
            data['attack_cat'] = data['attack_cat'].fillna('Benign')

            # update column dtypes for ports and clean data
            dropped_ports = ['-', '0x20205321']
            data = data[~data['sport'].isin(dropped_ports)]
            data = data[~data['dsport'].isin(dropped_ports)]

            data.reset_index(drop = True, inplace = True)

            # convert hexadecimal ports into base 10 before column conversion
            hex_ports = {
```

[9]https://github.com/SRI-CSL/trinity-packet

```
36                  '0x000b': 11,
37                  '0x000c': 12,
38                  '0xc0a8': 49320,
39                  '0xcc09': 52233,
40              }
41              # replace the hex ports
42              data['sport'] = data['sport'].replace(hex_ports)
43              data['dsport'] = data['dsport'].replace(hex_ports)
44
45              data['sport'] = data['sport'].astype(np.int32)
46              data['dsport'] = data['dsport'].astype(np.int32)
47
48              nflows += unsw_nb15_flow_processor(dataset, data, flow_labels, nflows)
49          else:
50              assert ('Unknown flow preprocessor attribute.')
51
52          sys.stdout.write('done in {:0.2f} seconds.\n'.format(time.time() - flow_start_time))
53          sys.stdout.flush()
54
55      # since the times are often at coarser resolution than the microsecond durations, sort
56      # by end time the last (start_time, end_time) that satisfies the constraint
57      # start_time <= timestamp <= end_time is the correct flow
58      for hash_tuple in flow_labels:
59          flow_labels[hash_tuple].sort(key = lambda x: x[1])
60
61      # save the hash filename
62      hash_filename = '{}/flow_labels_hash.pkl'.format(dataset.temp_directory)
63      with open(hash_filename, 'wb') as fd:
64          pickle.dump(flow_labels, fd, protocol = pickle.HIGHEST_PROTOCOL)
```

## A.2 UNSW-NB15 Specific Functions

When looking at flows in the UNSW-NB125 data, we only consider flows corresponding to the TCP and UDP protocols. We read the UNSW-NB15 spreadsheets into a pandas dataframe and focus on the following columns: srcip, sport, dstip, dsport, proto, Stime, Ltime, attack_cat. We return a running count of the total number of flows processed thus far. Note, we need to provide the dictionary and number of flows as arguments/return values because the flow-level datasets are typically split into multiple spreadsheets based on attack and/or time of day. The timestamps given with the flows are already in Coordinated Universal Time, and additionally counting up from the Linux epoch.

```
1      def unsw_nb15_flow_processor(dataset, data, flow_labels, previous_flows):
2      """
3      Process a given flow from UNSW-NB15 and add hashes to the flow labels. Returns the
4      number of processed flows.
5
6      @param dataset: ContextPCAPDataset variable with category mapping attribute
7      @param data: pandas dataframe with relevant flow information
8      @param flow_labels: a dictionary of flow hashes to start_time, end_time, category, etc.
9      @param previous_flows: number of previous flows seen
10     """
11     # count the number of flows processed
12     nflows = 0
13
14     # remove leading and trailing whitespace from column names
15     data.columns = data.columns.str.strip()
16
17     # go through every flow in the CSV file
18     for (src_ip, src_port, dest_ip, dest_port, protocol, start_time, end_time, category) in
           zip(
19         data['srcip'],
20         data['sport'],
21         data['dstip'],
22         data['dsport'],
23         data['proto'],
24         data['Stime'],
```

```
25            data['Ltime'],
26            data['attack_cat'],
27        ):
28            # TCP is protocol 6 and UDP is protocol 17 by the Internet Assigned Numbers
                  Authority (IANA)
29            if not protocol == 'tcp' and not protocol == 'udp': continue
30            protocol = protocol.upper()
31
32            # remove white space around category and add 'Benign' for empty attacks (benign
                  traffic)
33            category = category.strip()
34            if not len(category): category = 'Benign'
35
36            # update the category name if a category mapping exists
37            if hasattr(dataset, 'category_mapping'):
38                category = dataset.category_mapping[category]
39
40            # precision is zero second for the UNSW-NB15 datasets since the start and end
41            # time are both given (in seconds)
42            precision = 0
43
44            # create a list of this hash tuple if it doesn't already exist
45            hash_tuple = (src_ip, src_port, dest_ip, dest_port, protocol)
46            if not hash_tuple in flow_labels:
47                flow_labels[hash_tuple] = []
48            flow_labels[hash_tuple].append((start_time, end_time, precision, category,
                  previous_flows + nflows))
49            # since these flows are bidirectional, we also need to hash the reverse flow
                  information
50            # when iterating through dpkt, it will give (src_ip, src_port, dest_ip, dest_port,
                  protocol)
51            # but the CSV file only considers the src_ip as the initiator of the connection,
                  will miss
52            # all response packets
53            hash_tuple = (dest_ip, dest_port, src_ip, src_port, protocol)
54            if not hash_tuple in flow_labels:
55                flow_labels[hash_tuple] = []
56            flow_labels[hash_tuple].append((start_time, end_time, precision, category,
                  previous_flows + nflows))
57
58            # increment the number of flows seen
59            nflows += 1
60
61        return nflows
```

### A.3 CIC-IDS-2017 Specific Functions

One of the issues with converting the CIC-IDS-2017 flow-level data into packet-level data is the timestamps associated with the flows. These timestamps are in a string format both with and without second precision. This code takes a timestamp in the CIC-IDS-2017 string format and converts it into a timestamp in Coordinated Universal Time used by various PCAP capturing systems. We also return the precision of the time given. PCAP data has significantly higher time precision than these flow-level datasets. Thus, we keep track of the precision of the flows to provide lower and upper bounds for their actual start and end times.

```
1  def cic_ids_time_parser(str_time):
2      """
3      Time parser specific to the CIC-IDS-2017 dataset. Different datasets require different
4      parsing functions based on the time format. For the CIC-IDS-2017 dataset, attack
5      times are given in the New Brunswick timezone GMT-3 during the summer months. AM/PM
6      are not given but since all traffic occurs between 9am and 5pm, we add 12 to
7      afternoon hours.
8
9      @param str_time: the string representation of the time.
10     """
11     # get the date and time of the attack
```

```
12      date, time = str_time.split()
13      day, month, year = date.split('/')
14
15      # determine if seconds are given or not
16      if time.count(':') == 1:
17          hour, minute = time.split(':')
18          second, precision_in_seconds = 0, 60
19      else:
20          hour, minute, second = time.split(':')
21          precision_in_seconds = 1
22
23      # if the hour is in the afternoon (less than 8pm since some attacks
24      # begin in the 8th hours), add 12 to convert to military time
25      hour = int(hour)
26      if hour < 8: hour += 12
27      # add three to the hour to get the time in GMT format to match the
28      # times in dpkt pcap format. current times are in New Brunswick time
29      # in summer months which has GMT-3. we do not need to worry about the
30      # day changing since the last attacks occurs at 5pm ADT (8pm GMT)
31      hour += 3
32      # convert into a timestamp to get the float value for return
33      timestamp = datetime(
34          day = int(day),
35          month = int(month),
36          year = int(year),
37          hour = int(hour),
38          minute = int(minute),
39          second = int(second),
40          tzinfo = timezone.utc,
41      )
42
43      # convert to a float value and return the precision in seconds
44      return timestamp.timestamp(), precision_in_seconds
```

When looking at flows in the CIC-IDS-2017 data, we only consider flows corresponding to TCP and UDP. We read the CIC-IDS-2017 spreadsheets into a pandas dataframe and focus on the following columns: Source IP, Source Port, Destination IP, Destination Port, Protocol, Timestamp, Flow Duration, and Label. We return a running count of the total number of flows processed thus far. Note, we need to provide the dictionary and number of flows as arguments/return values because the flow-level datasets are typically split into multiple spreadsheets based on attack and/or time of day.

```
1   def cic_ids_flow_processor(dataset, data, flow_labels, previous_flows):
2       """
3       Process a given flow from CIC-IDS and add hashes to the flow labels. Returns the number
4       of processed flows.
5
6       @param dataset: ContextPCAPDataset variable with category mapping attribute
7       @param data: pandas dataframe with relevant flow information
8       @param flow_labels: a dictionary of flow hashes to start_time, end_time, category, etc.
9       @param previous_flows: number of previous flows seen
10      """
11      # count the number of flows processed
12      nflows = 0
13
14      # remove leading and trailing whitespace from column names
15      data.columns = data.columns.str.strip()
16
17      # go through every flow in the CSV file
18      for (src_ip, src_port, dest_ip, dest_port, protocol, timestamp, duration, category) in
              zip(
19          data['Source IP'],
20          data['Source Port'],
21          data['Destination IP'],
22          data['Destination Port'],
23          data['Protocol'],
24          data['Timestamp'],
```

```
25          data['Flow Duration'],
26          data['Label']
27      ):
28          # TCP is protocol 6 and UDP is protocol 17 by the Internet Assigned Numbers
                Authority (IANA)
29          if not protocol == 6 and not protocol == 17: continue
30          if protocol == 6: protocol = 'TCP'
31          else: protocol = 'UDP'
32
33          # get the start time as an integer conditioned on whether or not seconds are
                included
34          start_time, precision = cic_ids_time_parser(timestamp)
35          # CIC-IDS-2017 flow durations in microseconds (cannot add since the integral values
                are milliseconds)
36          end_time = (datetime.fromtimestamp(start_time) + timedelta(microseconds = int(
                duration))).timestamp()
37
38          # update the category name if a category mapping exists
39          if hasattr(dataset, 'category_mapping'):
40              category = dataset.category_mapping[category]
41
42          # create a list of this hash tuple if it doesn't already exist
43          hash_tuple = (src_ip, src_port, dest_ip, dest_port, protocol)
44          if not hash_tuple in flow_labels:
45              flow_labels[hash_tuple] = []
46          flow_labels[hash_tuple].append((start_time, end_time, precision, category,
                previous_flows + nflows))
47          # since these flows are bidirectional, we also need to hash the reverse flow
                information
48          # when iterating through dpkt, it will give (src_ip, src_port, dest_ip, dest_port,
                protocol)
49          # but the CSV file only considers the src_ip as the initiator of the connection,
                will miss
50          # all response packets
51          hash_tuple = (dest_ip, dest_port, src_ip, src_port, protocol)
52          if not hash_tuple in flow_labels:
53              flow_labels[hash_tuple] = []
54          flow_labels[hash_tuple].append((start_time, end_time, precision, category,
                previous_flows + nflows))
55
56          # increment the number of flows seen
57          nflows += 1
58
59      return nflows
```

### A.4   Matching Packet Data with Flow Labels

The following function takes a pcap file and the flow labels dictionaries generated in the previous sections and labels the packets themselves. There are a few main points to consider when constructing this function. The TCP flags are presented in a single integer value requiring us to do bit-wise operations to see which flags are set for each packet. For packets following the UDP protocols, we simply zero out those excluded values. We then identify the list of possible flows that match based on the five-tuple and its reciprocal:

$$(\text{src\_ip}, \text{src\_port}, \text{dest\_ip}, \text{dest\_port}, \text{protocol})$$
$$(\text{dest\_ip}, \text{dest\_port}, \text{src\_ip}, \text{src\_port}, \text{protocol})$$

Flow-level data constructors use a small set number of IP addresses and port numbers to more easily identify labels. However, this causes problems when assigning packets to flows for labeling. Thus for each packet's five-tuple, we iterate over all matching flows and identify those valid given the start and end times recorded previously. We note, however, that the precision of the flow-level datasets is coarser than the PCAP files provided. Thus, we match a packet to any flow where the packet's timestamp falls after the start time but before the end time plus the (im-)precision for that particular flow. There are occasionally multiple flows that will match with a given packet. This often happens (in these academic datasets) in denial-of-service attacks,

for example, where flow durations can be quite short. If the flow categories do not match, we simply drop the packet from analysis to not pollute the data. At this point, we have the information needed to store into a dataframe for input into the SAFE-NID program. Note, a real-world implementation of SAFE-NID would not require this burdensome processing since there would be no labels needed.

```python
def inet_to_str(inet):
    """
    Convert an inet object to a string representation.

    @param inet: inet network address
    """
    # return either IPv4 or IPv6 address
    try:
        return socket.inet_ntop(socket.AF_INET, inet)
    except ValueError:
        return socket.inet_ntop(socket.AF_INET6, inet)

def read_pcap_data(pcap_file, flow_labels):
    """
    Given a pcap class, read the pcap data packet by packet.

    @param pcap: type PCAPFile that contains filenames and types
    @param flow_labels: labels for each flow identified by tuple (src ip, src port, dest ip,
    dest port)
    """
    # store all packets into an array to save into a dataframe later
    packets = []

    # read this PCAP file and return the TCP/UDP packets
    with open(pcap_file.filename, 'rb') as pcap_fd:
        pcap_start_time = time.time()
        sys.stdout.write('Reading PCAP file {}...'.format(pcap_file.filename))
        # read either PCAP or PCAP next generation formats
        if pcap_file.pcap_type == 'PCAP':
            pcap = dpkt.pcap.Reader(pcap_fd)
        elif pcap_file.pcap_type == 'PCAPNG':
            pcap = dpkt.pcapng.Reader(pcap_fd)

        npackets = 0
        ndropped = 0
        # read each packet in the pcap file
        for timestamp, packet in pcap:
            # Linux cooked capture
            if pcap.datalink() == dpkt.pcap.DLT_LINUX_SLL:
                eth = dpkt.sll.SLL(packet)
            # ethernet format
            else:
                eth = dpkt.ethernet.Ethernet(packet)

            # skip any non IP IPv6 packets
            if not isinstance(eth.data, dpkt.ip.IP) and not isinstance(eth.data, dpkt.ip6.
                IP6):
                continue

            # get the IP data
            ip = eth.data
            # get relevant fields from the IP packet header (convert to string
            # representation)
            src_ip = inet_to_str(ip.src)
            dest_ip = inet_to_str(ip.dst)
            # get specific attributes from the header based on the protocol type (IP v IPv6)
            if isinstance(eth.data, dpkt.ip.IP):
                ttl = ip.ttl
                total_length = ip.len
                internet_layer_protocol = 'IPv4'
            elif isinstance(eth.data, dpkt.ip6.IP6):
                ttl = ip.hlim
                total_length = ip.plen
```

```
63                       internet_layer_protocol = 'IPv6'
64                   else:
65                       assert ('Unknown internet layer protocol.')
66
67               # skip any non UDP and TCP protocols (n.b., get_proto has a bug and does not
68               # recognize certain protocols, so we need to skip the others here)
69               if not isinstance(ip.data, dpkt.tcp.TCP) and not isinstance(ip.data, dpkt.udp.
                     UDP):
70                   continue
71
72               # get the protocol name
73               transport_layer_protocol = ip.get_proto(ip.p).__name__
74               transport_layer_protocol = transport_layer_protocol.upper()
75
76               # count the number of packets before any drops
77               npackets += 1
78
79               # get the data from the transport layer and convert to bytes
80               transport = ip.data
81               data = transport.data
82
83               # get the source and destination ports
84               src_port = transport.sport
85               dest_port = transport.dport
86
87               # get relevant header information for TCP
88               if isinstance(ip.data, dpkt.tcp.TCP):
89                   # need to divide by 2 raised to the bit location to get 0 and 1 values
90                   cwr_flag = (transport.flags & dpkt.tcp.TH_CWR) // 128
91                   ece_flag = (transport.flags & dpkt.tcp.TH_ECE) // 64
92                   urg_flag = (transport.flags & dpkt.tcp.TH_URG) // 32
93                   ack_flag = (transport.flags & dpkt.tcp.TH_ACK) // 16
94                   psh_flag = (transport.flags & dpkt.tcp.TH_PUSH) // 8
95                   rst_flag = (transport.flags & dpkt.tcp.TH_RST) // 4
96                   syn_flag = (transport.flags & dpkt.tcp.TH_SYN) // 2
97                   fin_flag = (transport.flags & dpkt.tcp.TH_FIN)
98               # all flags have zero value for UDP
99               else:
100                  cwr_flag = 0
101                  ece_flag = 0
102                  urg_flag = 0
103                  ack_flag = 0
104                  psh_flag = 0
105                  rst_flag = 0
106                  syn_flag = 0
107                  fin_flag = 0
108
109              # get the label for this packet
110              hash_tuple = (src_ip, src_port, dest_ip, dest_port, transport_layer_protocol)
111              # skip tuples that do not appear in the flow data
112              # for the CIC-IDS-2017 dataset, this will include all IPv6 traffic
113              if not hash_tuple in flow_labels:
114                  ndropped += 1
115                  continue
116
117              # go through possible hashed label values until identifying the proper flow
118              # based on the flow's start and end times. there can be multiple matches
119              # since the start time resolution is in seconds and the end time resolution is
120              # in microseconds so find the last flow that fits the conditions (must be
121              # the correct one)
122              packet_category = None
123              packet_flow_id = None
124              # the start time precision is only to the second (at best), we allow a small
125              # buffer for flows to be considered. if all flows in the buffer have the
126              # same category, the packet receives that category label
127              buffer_categories = set()
128              buffer_flow_ids = set()
```

```
129                 for (start_time, end_time, precision, category, flow_id) in flow_labels[
                        hash_tuple]:
130                     # consider all possible flows that can fall in this time frame
131                     if start_time <= timestamp and timestamp <= end_time + precision:
132                         buffer_categories.add(category)
133                         buffer_flow_ids.add(flow_id)
134
135                 # continue if there are overlapping flows with different category labels
136                 if len(buffer_categories) == 1:
137                     packet_category = list(buffer_categories)[0]
138                     # take the first flow ID (not a perfect method)
139                     packet_flow_id = list(buffer_flow_ids)[0]
140                 else:
141                     ndropped += 1
142                     continue
143
144                 # only Benign category is labeled as non-attack
145                 packet_label = 0 if packet_category == 'Benign' else 1
146
147                 # get the hex representation of the payload
148                 if isinstance(data, bytes):
149                     payload = data.hex()
150                 else:
151                     payload = (data.pack()).hex()
152
153                 # save the payload length (for pruning later)
154                 payload_length = len(payload)
155
156                 packets.append([
157                     src_ip,
158                     src_port,
159                     dest_ip,
160                     dest_port,
161                     timestamp,
162                     internet_layer_protocol,
163                     transport_layer_protocol,
164                     ttl,
165                     total_length,
166                     cwr_flag,
167                     ece_flag,
168                     urg_flag,
169                     ack_flag,
170                     psh_flag,
171                     rst_flag,
172                     syn_flag,
173                     fin_flag,
174                     payload,
175                     payload_length,
176                     packet_flow_id,
177                     packet_category,
178                     packet_label,
179                 ])
180
181         sys.stdout.write('read {} packets (dropped: {}) in {:0.2f} seconds.\n'.format(
                    npackets, ndropped, time.time() - pcap_start_time))
182         sys.stdout.flush()
183
184     return packets
```

# B   Complete Out-of-Distribution Results

Table 12: *FTP-Patator out-of-distribution results.* The transformer architecture with either the Gaussian kernel density or normalizing flows safeguard achieves very high AU ROC scores > 0.9936. The FNN model performs better than the CNN on this data, but still is significantly worse than when using a transformer architecture.

| Architecture | Layer | AU ROC (↑) | TPR (TNR 95%) (↑) | TPR (TNR 85%) (↑) |
|---|---|---|---|---|
| **FTP-Patator** | | | | |
| **Maximum Softmax Probability** | | | | |
| CNN | Output | 0.7931 (±0.1534) | 15.45% (±15.48%) | 64.19% (±37.97%) |
| FNN | Output | 0.6149 (±0.0893) | 0.12% (±0.37%) | 9.89% (±11.39%) |
| Transformer | Output | 0.4034 (±0.2232) | 0.21% (±0.54%) | 14.61% (±13.37%) |
| **Energy-Based** | | | | |
| CNN | Output | 0.7628 (±0.1657) | 12.51% (±12.78%) | 52.82% (±32.85%) |
| FNN | Output | 0.6060 (±0.1500) | 0.19% (±0.56%) | 16.55% (±19.86%) |
| Transformer | Output | 0.4133 (±0.1955) | 0.21% (±0.51%) | 15.98% (±17.58%) |
| **Gaussian Kernel Density** | | | | |
| CNN | dense1 | 0.7563 (±0.2059) | 0.53% (±0.51%) | 57.66% (±26.93%) |
| CNN | dense2 | 0.8207 (±0.0684) | 4.09% (±2.69%) | 51.64% (±30.72%) |
| CNN | dense3 | 0.8297 (±0.1157) | 26.47% (±24.29%) | 58.92% (±29.76%) |
| FNN | linear3 | 0.8473 (±0.0259) | 3.55% (±8.46%) | 59.20% (±19.59%) |
| FNN | linear4 | 0.7511 (±0.0406) | 2.56% (±4.37%) | 19.70% (±15.77%) |
| FNN | linear5 | 0.7585 (±0.0591) | 0.37% (±0.47%) | 24.53% (±17.33%) |
| Transformer | linear1 | **0.9950 (±0.0018)** | **100.00% (±0.00%)** | **100.00% (±0.00%)** |
| Transformer | linear2 | 0.9717 (±0.0083) | 89.63% (±16.73%) | **100.00% (±0.00%)** |
| Transformer | linear3 | 0.9255 (±0.0406) | 44.22% (±33.00%) | 89.51% (±16.49%) |
| **Normalizing Flows** | | | | |
| CNN | dense1 | 0.7442 (±0.2052) | 0.50% (±0.72%) | 51.70% (±35.34%) |
| CNN | dense2 | 0.8058 (±0.1224) | 4.53% (±5.42%) | 48.96% (±31.88%) |
| CNN | dense3 | 0.7998 (±0.1178) | 15.46% (±14.41%) | 56.90% (±29.89%) |
| FNN | linear3 | 0.8640 (±0.0152) | 6.88% (±12.78%) | 61.51% (±14.50%) |
| FNN | linear4 | 0.7917 (±0.0389) | 3.37% (±8.52%) | 29.55% (±14.60%) |
| FNN | linear5 | 0.7732 (±0.0580) | 7.25% (±7.74%) | 38.37% (±13.60%) |
| Transformer | linear1 | 0.9936 (±0.0014) | **100.00% (±0.00%)** | **100.00% (±0.00%)** |
| Transformer | linear2 | 0.9726 (±0.0064) | 92.71% (±13.52%) | **100.00% (±0.00%)** |
| Transformer | linear3 | 0.9480 (±0.0199) | 58.83% (±26.97%) | 97.50% (±4.21%) |

Table 13: *Infiltration out-of-distribution results.* The transformer with normalizing flows safeguard performs worse here than with the other two zero-day exploits. However, is still outperforms baseline methods on AU ROC. The CNN architecture performs well here compared to FTP-Patator and SSH-Patator results.

| Architecture | Layer | AU ROC (↑) | TPR (TNR 95%) (↑) | TPR (TNR 85%) (↑) |
|---|---|---|---|---|
| | | Infiltration | | |
| | | Maximum Softmax Probability | | |
| CNN | Output | 0.6611 (±0.1897) | 26.16% (±20.54%) | 53.95% (±23.83%) |
| FNN | Output | 0.7676 (±0.0567) | 12.39% (±11.58%) | 47.52% (±16.07%) |
| Transformer | Output | 0.3167 (±0.2055) | 3.37% (±6.66%) | 9.56% (±19.88%) |
| | | Energy-Based | | |
| CNN | Output | 0.6744 (±0.1859) | 25.33% (±21.43%) | 53.56% (±26.27%) |
| FNN | Output | 0.7592 (±0.0750) | 11.41% (±10.28%) | 47.91% (±15.66%) |
| Transformer | Output | 0.3037 (±0.2291) | 4.33% (±11.43%) | 10.13% (±23.19%) |
| | | Gaussian Kernel Density | | |
| CNN | dense1 | 0.9581 (±0.0062) | **83.74% (±10.23%)** | 98.70% (±0.53%) |
| CNN | dense2 | 0.9511 (±0.0218) | 75.73% (±19.16%) | 96.55% (±2.93%) |
| CNN | dense3 | 0.9622 (±0.0217) | 79.11% (±15.19%) | 96.29% (±2.99%) |
| FNN | linear3 | 0.8740 (±0.0337) | 27.72% (±13.36%) | 67.78% (±13.74%) |
| FNN | linear4 | 0.8748 (±0.0405) | 41.70% (±15.30%) | 61.45% (±16.68%) |
| FNN | linear5 | 0.8345 (±0.0439) | 29.58% (±19.39%) | 47.85% (±18.80%) |
| Transformer | linear1 | 0.9455 (±0.0248) | 62.58% (±20.95%) | 93.79% (±10.34%) |
| Transformer | linear2 | 0.9552 (±0.0318) | 71.26% (±28.99%) | 94.93% (±13.73%) |
| Transformer | linear3 | 0.8996 (±0.0580) | 35.48% (±28.28%) | 75.45% (±25.56%) |
| | | Normalizing Flows | | |
| CNN | dense1 | 0.9464 (±0.0061) | 67.72% (±15.48%) | 98.90% (±0.05%) |
| CNN | dense2 | 0.9519 (±0.0057) | 72.07% (±18.42%) | 98.91% (±0.06%) |
| CNN | dense3 | 0.9638 (±0.0153) | 77.08% (±17.79%) | 98.22% (±0.78%) |
| FNN | linear3 | 0.8655 (±0.0115) | 5.89% (±4.88%) | 66.73% (±11.63%) |
| FNN | linear4 | 0.8820 (±0.0300) | 30.41% (±15.53%) | 67.72% (±14.22%) |
| FNN | linear5 | 0.8530 (±0.0385) | 29.11% (±18.50%) | 59.60% (±14.84%) |
| Transformer | linear1 | 0.9555 (±0.0146) | 63.20% (±24.30%) | **99.64% (±0.62%)** |
| Transformer | linear2 | **0.9668 (±0.0182)** | 77.89% (±23.41%) | 99.18% (±2.25%) |
| Transformer | linear3 | 0.9361 (±0.0455) | 56.13% (±31.83%) | 89.54% (±15.68%) |

Table 14: *SSH-Patator out-of-distribution results.* The transformer classifier with normalizing flows safeguard outperforms all existing methods with a very high AU ROC of 0.9901.

| Architecture | Layer | AU ROC (↑) | TPR (TNR 95%) (↑) | TPR (TNR 85%) (↑) |
|---|---|---|---|---|
| **SSH-Patator** | | | | |
| **Maximum Softmax Probability** | | | | |
| CNN | Output | 0.7821 (±0.0511) | 17.19% (±10.35%) | 66.61% (±15.90%) |
| FNN | Output | 0.6568 (±0.0727) | 1.53% (±0.87%) | 31.75% (±11.64%) |
| Transformer | Output | 0.1924 (±0.1103) | 0.20% (±0.35%) | 2.24% (±2.45%) |
| **Energy-Based** | | | | |
| CNN | Output | 0.7807 (±0.0701) | 18.90% (±9.81%) | 64.77% (±14.63%) |
| FNN | Output | 0.5932 (±0.0896) | 1.38% (±0.92%) | 23.38% (±12.69%) |
| Transformer | Output | 0.1833 (±0.1142) | 0.19% (±0.34%) | 2.12% (±2.57%) |
| **Gaussian Kernel Density** | | | | |
| CNN | dense1 | 0.6158 (±0.1096) | 7.14% (±1.62%) | 27.81% (±13.33%) |
| CNN | dense2 | 0.6993 (±0.1041) | 11.45% (±7.93%) | 39.42% (±11.65%) |
| CNN | dense3 | 0.7254 (±0.1269) | 23.72% (±10.20%) | 46.08% (±12.92%) |
| FNN | linear3 | 0.8967 (±0.0100) | 26.14% (±4.38%) | 77.99% (±5.01%) |
| FNN | linear4 | 0.8381 (±0.0169) | 19.34% (±3.42%) | 52.81% (±6.62%) |
| FNN | linear5 | 0.8298 (±0.0265) | 18.36% (±3.01%) | 56.14% (±7.42%) |
| Transformer | linear1 | 0.9821 (±0.0037) | 94.90% (±2.61%) | 99.05% (±1.09%) |
| Transformer | linear2 | 0.9629 (±0.0048) | 76.63% (±9.36%) | **100.00% (±0.00%)** |
| Transformer | linear3 | 0.9097 (±0.0313) | 27.03% (±14.36%) | 86.99% (±12.27%) |
| **Normalizing Flows** | | | | |
| CNN | dense1 | 0.5739 (±0.1273) | 6.90% (±0.51%) | 18.94% (±6.43%) |
| CNN | dense2 | 0.7086 (±0.1117) | 11.23% (±4.00%) | 39.33% (±13.11%) |
| CNN | dense3 | 0.7010 (±0.1336) | 21.97% (±11.52%) | 44.46% (±15.20%) |
| FNN | linear3 | 0.9119 (±0.0062) | 30.81% (±3.04%) | 84.98% (±3.99%) |
| FNN | linear4 | 0.8810 (±0.0144) | 30.83% (±3.27%) | 70.58% (±6.78%) |
| FNN | linear5 | 0.8596 (±0.0312) | 30.49% (±3.33%) | 67.67% (±8.18%) |
| Transformer | linear1 | **0.9901 (±0.0007)** | **100.00% (±0.00%)** | **100.00% (±0.00%)** |
| Transformer | linear2 | 0.9668 (±0.0062) | 84.62% (±12.51%) | **100.00% (±0.00%)** |
| Transformer | linear3 | 0.9323 (±0.0180) | 45.45% (±13.92%) | 92.96% (±6.04%) |

Table 15: *CIC-IDS-2017 out-of-distribution results.* The highest performing combination for detecting the out-of-distribution samples is the the transformer descriminative classifier with the normalizing flows safeguard.

| Architecture | Layer | AU ROC (↑) | TPR (TNR 95%) (↑) | TPR (TNR 85%) (↑) |
|---|---|---|---|---|
| **CIC-IDS-2017** | | | | |
| **Maximum Softmax Probability** | | | | |
| CNN | Output | 0.7818 (±0.0254) | 36.43% (±4.48%) | 63.02% (±4.47%) |
| FNN | Output | 0.7187 (±0.0289) | 23.25% (±3.21%) | 50.69% (±3.18%) |
| Transformer | Output | 0.6731 (±0.0737) | 23.41% (±8.93%) | 38.37% (±14.72%) |
| **Energy-Based** | | | | |
| CNN | Output | 0.7589 (±0.0436) | 36.79% (±6.40%) | 60.53% (±3.58%) |
| FNN | Output | 0.7114 (±0.0498) | 22.90% (±3.63%) | 49.71% (±3.14%) |
| Transformer | Output | 0.6419 (±0.0966) | 21.88% (±9.79%) | 35.48% (±14.45%) |
| **Gaussian Kernel Density** | | | | |
| CNN | dense1 | 0.8814 (±0.0144) | 49.93% (±4.44%) | 75.21% (±3.38%) |
| CNN | dense2 | 0.8981 (±0.0170) | 55.39% (±5.94%) | 77.16% (±5.28%) |
| CNN | dense3 | 0.8580 (±0.0394) | 55.26% (±4.30%) | 65.82% (±5.45%) |
| FNN | linear3 | 0.6986 (±0.0464) | 25.80% (±5.13%) | 40.43% (±6.12%) |
| FNN | linear4 | 0.7343 (±0.0597) | 34.96% (±4.17%) | 45.25% (±5.73%) |
| FNN | linear5 | 0.6801 (±0.0520) | 25.72% (±2.39%) | 35.15% (±3.33%) |
| Transformer | linear1 | 0.8323 (±0.0649) | 50.61% (±8.24%) | 64.29% (±10.40%) |
| Transformer | linear2 | 0.8753 (±0.0552) | 58.05% (±11.80%) | 73.07% (±11.11%) |
| Transformer | linear3 | 0.7893 (±0.0541) | 42.14% (±6.95%) | 53.56% (±9.77%) |
| **Normalizing Flows** | | | | |
| CNN | dense1 | 0.8744 (±0.0121) | 30.50% (±3.95%) | 71.26% (±5.71%) |
| CNN | dense2 | 0.8960 (±0.0151) | 52.88% (±3.63%) | 76.19% (±5.61%) |
| CNN | dense3 | 0.8384 (±0.0594) | 53.77% (±5.61%) | 67.10% (±8.80%) |
| FNN | linear3 | 0.8130 (±0.0226) | 27.27% (±6.58%) | 50.10% (±2.03%) |
| FNN | linear4 | 0.7625 (±0.0210) | 35.61% (±1.12%) | 48.22% (±2.08%) |
| FNN | linear5 | 0.7139 (±0.0582) | 31.64% (±2.98%) | 42.69% (±5.90%) |
| Transformer | linear1 | 0.9011 (±0.0115) | 66.87% (±3.37%) | 76.70% (±2.31%) |
| Transformer | linear2 | **0.9259 (±0.0046)** | **73.01% (±2.10%)** | **81.55% (±1.11%)** |
| Transformer | linear3 | 0.8945 (±0.0085) | 63.15% (±4.94%) | 77.45% (±2.06%) |

Table 16: *UNSW-NB15 out-of-distribution results.* Although transformer with normalizing flows performs well, the best performer comes form the transformer with the Gaussian kernel density safeguard.

| Architecture | Layer | AU ROC (↑) | TPR (TNR 95%) (↑) | TPR (TNR 85%) (↑) |
|---|---|---|---|---|
| | | **UNSW-NB15** | | |
| | | **Maximum Softmax Probability** | | |
| CNN | Output | 0.5590 (±0.0063) | 34.87% (±2.45%) | 44.47% (±3.19%) |
| FNN | Output | 0.7349 (±0.0417) | 25.98% (±2.04%) | 43.43% (±5.41%) |
| Transformer | Output | 0.6255 (±0.1033) | 24.21% (±11.24%) | 40.14% (±14.78%) |
| | | **Energy-Based** | | |
| CNN | Output | 0.5381 (±0.0480) | 32.24% (±5.82%) | 41.78% (±8.91%) |
| FNN | Output | 0.7177 (±0.0977) | 26.62% (±2.73%) | 44.61% (±8.65%) |
| Transformer | Output | 0.6153 (±0.1003) | 24.26% (±11.29%) | 39.55% (±14.19%) |
| | | **Gaussian Kernel Density** | | |
| CNN | dense1 | 0.9137 (±0.0208) | 53.07% (±2.51%) | 79.71% (±7.07%) |
| CNN | dense2 | 0.9263 (±0.0139) | 61.61% (±4.44%) | 83.14% (±4.44%) |
| CNN | dense3 | 0.9217 (±0.0118) | 62.52% (±3.50%) | 81.29% (±4.04%) |
| FNN | linear3 | 0.8600 (±0.0178) | 27.30% (±4.04%) | 65.76% (±6.38%) |
| FNN | linear4 | 0.9079 (±0.0109) | 56.06% (±5.32%) | 80.31% (±2.68%) |
| FNN | linear5 | 0.8759 (±0.0235) | 43.55% (±9.04%) | 71.70% (±6.78%) |
| Transformer | linear1 | 0.8762 (±0.0227) | 43.21% (±6.52%) | 64.55% (±8.58%) |
| Transformer | linear2 | **0.9636 (±0.0126)** | **79.23% (±8.25%)** | **95.73% (±2.68%)** |
| Transformer | linear3 | 0.9550 (±0.0094) | 71.49% (±8.26%) | 95.27% (±1.73%) |
| | | **Normalizing Flows** | | |
| CNN | dense1 | 0.9071 (±0.0047) | 48.95% (±1.34%) | 74.30% (±2.58%) |
| CNN | dense2 | 0.9263 (±0.0063) | 55.98% (±1.29%) | 83.03% (±2.72%) |
| CNN | dense3 | 0.8982 (±0.0130) | 58.78% (±1.38%) | 72.69% (±3.38%) |
| FNN | linear3 | 0.8567 (±0.0137) | 22.42% (±3.62%) | 62.69% (±6.80%) |
| FNN | linear4 | 0.8963 (±0.0112) | 45.10% (±8.46%) | 77.66% (±3.28%) |
| FNN | linear5 | 0.8549 (±0.0219) | 36.02% (±7.18%) | 64.46% (±5.96%) |
| Transformer | linear1 | 0.9263 (±0.0071) | 77.83% (±0.39%) | 81.84% (±1.61%) |
| Transformer | linear2 | 0.9536 (±0.0072) | 77.96% (±3.03%) | 91.07% (±2.59%) |
| Transformer | linear3 | 0.9583 (±0.0090) | 77.71% (±4.49%) | 94.44% (±2.24%) |

# C ODIN Results

One critical component of our system is its ability to work on zero day exploits. Thus, OOD methods that require outlier exposure would fail in this set up. Furthermore, methods that require some level of parameter tuning generally fall out of scope for the same reason. Thus, we use the parameter-less version of energy-based detection. Nevertheless, we test ODIN (Liang et al., 2017a) as our model safeguard as well and provide those results here. Since we do not allow for parameter tuning, we use the default parameters ($T = 1000, \epsilon = 0.05$) from the `pytorch-ood` library (Kirchheim et al., 2022).

We find a large number of identical output values from the ODIN detector. Thus, the values for TPR (TNR 95%) or TPR (TNR 85%) are skewed low. That is, the threshold value encompasses a much larger percentage of true negatives (since in some cases over 20% of ID output values are identical). We include the values even still for comparison in Table 17. However, we note that this would make such a safeguard unusable in practice because the threshold would be too hard to tune.

Table 17: *ODIN results.* We use ODIN as a model safeguard with default parameters from the `pytorch-ood` library.

| Architecture | Layer | AU ROC (↑) | TPR (TNR 95%) (↑) | TPR (TNR 85%) (↑) |
|---|---|---|---|---|
| **FTP-Patator** | | | | |
| CNN | Output | 0.4881 (±0.2614) | 0.00% (±0.00%) | 0.05% (±0.13%) |
| FNN | Output | 0.9193 (±0.0265) | 0.00% (±0.00%) | 73.05% (±38.27%) |
| Transformer | Output | 0.5633 (±0.1748) | 0.00% (±0.00%) | 0.00% (±0.00%) |
| **Infiltration** | | | | |
| CNN | Output | 0.7131 (±0.1616) | 11.14% (±14.62%) | 29.59% (±30.75%) |
| FNN | Output | 0.5441 (±0.1439) | 0.00% (±0.00%) | 6.22% (±6.42%) |
| Transformer | Output | 0.6231 (±0.1990) | 0.00% (±0.00%) | 10.68% (±21.46%) |
| **SSH-Patator** | | | | |
| CNN | Output | 0.3665 (±0.2301) | 0.85% (±1.98%) | 4.56% (±3.79%) |
| FNN | Output | 0.7255 (±0.0555) | 0.00% (±0.00%) | 26.86% (±15.71%) |
| Transformer | Output | 0.7277 (±0.1070) | 2.65% (±7.94%) | 8.13% (±18.65%) |
| **CIC-IDS-2017** | | | | |
| CNN | Output | 0.4914 (±0.0251) | 0.72% (±2.15%) | 8.44% (±11.05%) |
| FNN | Output | 0.5687 (±0.0486) | 0.00% (±0.00%) | 7.28% (±7.35%) |
| Transformer | Output | 0.3072 (±0.0586) | 0.83% (±1.66%) | 7.11% (±5.73%) |
| **UNSW-NB15** | | | | |
| CNN | Output | 0.6321 (±0.0251) | 22.50% (±18.47%) | 24.91% (±20.40%) |
| FNN | Output | 0.4167 (±0.0565) | 0.00% (±0.00%) | 12.59% (±6.48%) |
| Transformer | Output | 0.4422 (±0.0694) | 1.84% (±3.87%) | 5.67% (±7.41%) |

