# OpenReview forum: "SAFE-NID: Self-Attention with Normalizing-Flow Encodings for Network Intrusion Detection"
_TMLR — Accepted by TMLR_

### Review · Reviewer_gPUF · 2024-11-15

**Summary Of Contributions:**

SAFE-NID introduces a encoder-only transformer model for packet-level intrusion detection, employs normalizing flows to assess uncertainty in classifications, and provides packet-level datasets to better simulate real-world attack scenarios.

**Audience:**

Yes

**Claims And Evidence:**

No

**Requested Changes:**

I am not an expert in this field, but my primary concern is the focus of the evaluation in this paper. In my view, the emphasis should be on the construction of the model safeguards system, rather than the proposed SAFE-NID. Additionally, all evaluations appear to be based solely on the baseline model without comparison to other methods, which significantly undermines the credibility of the assessment. I recommend the authors make the following improvements: 1. Focus on the newly proposed model safeguards system. 2. Provide evaluation comparisons with current SOTA methods or elaborate on the rationale for not doing so.

**Strengths And Weaknesses:**

Strengths:
- SAFE-NID demonstrates efficacy in the detection of novel attacks.
- The provision of extensive packet-level datasets is conducive to advancing further inquiry.
- The paper encompasses thorough experimental evaluations.
- The research provides in-depth insights into packet-level classification.

Weaknesses:
- The performance on new attacks may heavily rely on data diversity.
- The comparative methods appears to be unconvincing.

---

> ### Author Response · Authors · 2025-01-22
> **Response to Reviewer**
>
> We appreciate the reviewer’s thoughtful feedback and recognition of the strengths of our work, including its efficacy in detecting novel attacks, the contribution of packet-level datasets, and comprehensive experimental evaluations. Below, we address the weaknesses, requested changes, and other concerns raised in the review.
>
> ## **Weaknesses and Suggested Changes:**
>
> 1) **Focus on Model Safeguard System:** While SAFE-NID as a whole represents a novel framework, we acknowledge the need to emphasize its safeguard components (e.g., normalizing flows for uncertainty estimation) and their role in enhancing detection capabilities. In the revised paper, we emphasize on the safeguard system design and its contributions to improving detection reliability.
> 2) **Dependence on Data Diversity:** The reviewer raises an important point regarding SAFE-NID’s reliance on data diversity for performance on novel attacks. We emphasize that our OOD detection method does not lie on outlier exposure. So, we can detect novel packets we have not seen before in the training set. Some of these novel packets might be benign.
> 3) **Comparative SOTA methods in the experiments and evaluation:** To address this, we included comparisons with current state-of-the-art methods for out-of-distribution (OOD) detection and intrusion detection, such as an energy-based method. We note that our focus on zero-day exploits requires the use of “parameter free” models without the need for outlier exposure. Thus, for our energy-based comparison we use a temperature of 1.0. We performed ablation studies using ODIN (without outlier exposure and the default temperature values from the pytorch-ood library) but found the results poor. We include a small summary of this work in the appendix.
>
> We thank the reviewer for their constructive feedback, which will significantly enhance the quality and clarity of our work. We are confident that the proposed revisions will address the concerns raised while strongly focusing on SAFE-NID’s novel contributions and comparative performance. Thank you for your time and valuable suggestions.

---

> > ### Comment · Reviewer_gPUF · 2025-02-01
> > **Final Review**
> >
> > Thanks for the response, most of my concerns have been addressed during the discussion. Overall, although the framework proposed in this paper mostly adopts established methods, its evaluation demonstrates its effectiveness. I believe it holds certain value in the field of network intrusion detection.

---

### Review · Reviewer_DcTh · 2024-11-29

**Summary Of Contributions:**

The paper introduces SAFE-NID, a novel framework for network intrusion detection (NID) that leverages self-attention mechanisms and normalizing-flow encodings. SAFE-NID performs packet-level analysis, enabling real-time detection of anomalies and zero-day attacks without waiting for complete flow-level information.

**Audience:**

Yes

**Broader Impact Concerns:**

1. Network intrusion detection in packet level may contain sensitive information such as user payloads, metadata, or session identifiers. Without proper safeguards, this process could violate privacy laws or erode user trust. Including a discussion on compliance with data privacy regulations and ensuring transparency in how the system processes and stores network traffic would address these concerns.

2. Lack of discussion regarding practical implications of adopting this method for scalable real-time intrusion analysis.

**Claims And Evidence:**

Yes

**Requested Changes:**

1. How sensitive is SAFE-NID’s performance to hyperparameters in the normalizing flows, such as the number of flow blocks or the clamping value? Have ablation studies been conducted to identify the optimal settings for these parameters?
2. Testing SAFE-NID against adversarial examples designed to evade out-of-distribution (OOD) detection, such as perturbed packets or modified attack payloads will be great value addition to the paper.
3. How does SAFE-NID ensure robustness against feature collapse in deeper layers, particularly when processing high-redundancy payload data?
4. Mention the limitations of the method.

**Strengths And Weaknesses:**

> Strengths:
1. SAFE-NID operates at the packet level, enabling faster detection compared to traditional flow-level methods. This is relevant for real-world applications requiring low-latency responses.
2. The authors have conducted extensive experiments on two major datasets, CIC-IDS-2017 and UNSW-NB15, including evaluations for zero-day attacks and distribution shifts.
3. The release of packet-level versions of the UNSW-NB15 and CIC-IDS-2017 datasets provides a valuable resource for the research community, enabling further advancements in packet-level intrusion detection.

> Weakness:
1. The computational overhead introduced by normalizing flows and transformer models is not analyzed in detail. The scalability of the approach to high-throughput environments is lacking.
2. No evaluation of the system’s resistance to adversarial inputs, such as adversarially crafted packets.
3. The paper does not sufficiently compare SAFE-NID with other established OOD detection techniques beyond Gaussian kernel density and maximum softmax probability. Include comparisons with recent methods like ODIN, Deep Ensembles, etc.
4. While the paper outlines challenges in converting flow-level data to packet-level, it lacks clarity and reproducibility details in handling packet overlaps or mismatched flow timestamps.

---

> ### Author Response · Authors · 2025-01-22
> **Response to Reviewer (Part 1)**
>
> We sincerely thank the reviewer for their detailed and constructive feedback. We are pleased that the reviewer recognizes the relevance and novelty of SAFE-NID’s packet-level approach for real-time network intrusion detection. We appreciate the acknowledgment of our:
> 1) Low-latency packet-level detection mechanism, which addresses critical needs in real-world applications.
> 2) Comprehensive experimental evaluations on two widely used datasets (CIC-IDS-2017 and UNSW-NB15), including zero-day attacks and distribution shifts.
> 3) Contribution of packet-level versions of the datasets as a valuable resource for the research community.
>
> Below, we address the points raised, including strengths, weaknesses, requested changes, and broader impact concerns.
>
> ## **Weaknesses and Suggested Changes:**
> 1) **Computational Overhead and Scalability:** We agree that a detailed analysis of computational overhead and scalability is essential for high-throughput environments. We revised to include a quantitative breakdown of SAFE-NID’s computational requirements for key components, such as normalizing flows and transformer-based models, in Section 6.5 Latency Analysis. Additionally, we include a discussion on using potential optimization strategies, such as quantization, pruning, distillation, or lightweight alternatives, to reduce computational costs.
> 2) **Adversarial Inputs and Robustness:** The absence of an evaluation against adversarial inputs is a valid concern. We have focussed on detecting OOD inputs, and robustness to adversarially crafted inputs is left to a future study. In the considered application of network intrusion detection, the failure of ML models to detect new attacks can be hazardous, and our work is focused on that problem. In this case, the attacker might not be specifically targeting the ML model or even aware that an ML model-based detection system is in place. A new attack could make the ML model fail, and we use OOD detection to safeguard the model against this failure. That said, crafting adversarial inputs in the payload space is an inherently difficult task because of encryption and compression. Slight perturbations in the input space will create corruptions after decompression and decryptions preventing said packets from rendering. Additionally, many applications would simply reject these malformed packets.
> 3) **Comparative Analysis with Other OOD Techniques:** While we evaluated SAFE-NID against Gaussian Kernel Density and Maximum Softmax Probability, we recognize the importance of broader comparisons. We include a comparison with energy-based models in our paper, providing a more comprehensive evaluation. We note that our focus on zero-day exploits requires the use of “parameter free” models without the need for outlier exposure. Thus, for our energy-based comparison we use a temperature of 1.0. We performed ablation studies using ODIN (without outlier exposure and the default temperature values from the pytorch-ood library) but found the results poor. We include a small summary of this work in the appendix.
> 4) **Clarity on Packet-Level Data Handling:** To improve clarity and reproducibility, we have expanded the explanation of our methodology for converting flow-level data to packet-level format, including how we handle packet overlaps and mismatched flow timestamps in Section 4.2. We have also added additional explanations with code snippets in Appendix A. We will release the code with the final paper.
> 5) **Hyperparameter Sensitivity and Ablation Studies:** We appreciate the suggestion to explore hyperparameter sensitivity. In the revised paper, we added an ablation study to analyze the impact of key hyperparameters in normalizing flows, such as the number of flow blocks and clamping values (Section 6.6). In summary, the clamping value has little effect on the validation loss during training. There is generally a decrease in validation loss when increasing the number of blocks - however, with diminishing returns. We find that the validation loss generally plateaus around 20 blocks and the deeper networks are not worth the additional inference time.
> 6) **Robustness Against Feature Collapse:** The feature collapse in deeper layers is definitely an important challenge in OOD detection. We extended the Related Works section to include a discussion on this. We note that our approach considers not just the payload but the full packet (excluding fields that we found too informative in the dataset due to the limitations of how the data was collected). The Appendix includes the results from each combination of neural architecture and model safeguard. We note some feature collapse for the FNN and Transformer resulting in decreasing AU ROC for the last hidden layers. We have highlighted this in Section 6.5 in the main text.

---

> ### Author Response · Authors · 2025-01-22
> **Response to Reviewer (Part 2)**
>
> 7) **Limitations:** We include a discussion on limitations in the Conclusion section, such as computational overhead, potential scalability challenges, and areas requiring further development, such as adversarial robustness.
>
> **Broader Impact Concerns:** We agree that network intrusion detection may involve processing sensitive information. The revised paper discusses challenges and safeguards to protect user data privacy, including encryption of sensitive payloads, metadata minimization, and compliance with privacy laws like GDPR and CCPA (Section 6.7). We also emphasize enterprise networks, where collective security trumps individual privacy, as the most likely use case. We also address the practical implications of deploying SAFE-NID in scalable real-time environments, including resource requirements and optimizations for deployment on edge devices or cloud-based systems (Section 6.6). The emergence of low-cost and high-throughput specialized hardware for ML inference encourages the practical adoption of ML-based network intrusion detection, and several such methods have already been deployed, such as Cisco Secure Network Analysis, CrowdStrike ExtraHop, Darktrace, and Sangfor Cyber Command. Our approach provides a lightweight safeguard to detect OOD inputs to such ML models.
>
> We are grateful for the reviewer’s insightful feedback and are confident that the proposed revisions will address the concerns raised while enhancing the clarity and impact of our work. Thank you for your thoughtful suggestions and consideration.

---

### Review · Reviewer_jnMv · 2024-12-25

**Summary Of Contributions:**

This paper presents SAFE-NID, which utilizes deep neural networks for network intrusion detection at the packet level. SAFE-NID includes 2 components. The first is a deep neural network that performs the binary classification to classify between benign and malicious packets. In the second component, they consider the OOD setting to detect zero-day attacks and simulate the scenario with inputs from a different time. In the evaluation, they first show that when the test inputs are in-distribution, all model architectures, including CNN, feed-forward neural networks and Transformer, achieve nearly perfect detection accuracy. Then they show that when facing inputs that belong to the attack type not appearing in the training set, the performance drops dramatically, and the detection accuracy is close to 0 for certain attack types. Finally, they evaluate different OOD detection techniques, including maximum softmax probability, Gaussian Kernel Density, and Normalizing Flows, and they show that Normalizing Flows achieves decent detection performance across different settings.

**Audience:**

Yes

**Broader Impact Concerns:**

No concerns.

**Claims And Evidence:**

Yes

**Requested Changes:**

1. Re-organize the paper to highlight the novelty of the paper.

2. Add more discussion and analysis about each set of OOD detection experiments, see details in the Weaknesses section.

**Strengths And Weaknesses:**

Strengths:

1. To my knowledge, network intrusion detection in the OOD setting has not been well-studied in the literature, but this is an important topic for practical usage of these techniques.

2. This work conducts an extensive study on different attack types, OOD detection techniques, implementation of neural network architectures, etc., to demonstrate how each of these factors affects the performance.

Weaknesses:

1. One main weakness of this work is that the novelty is very limited from the machine learning perspective. The neural network architectures and OOD detection techniques evaluated in this paper all come from existing work. It is unclear what are some unique challenges that need to be considered when applying existing techniques to the new application in this work.

2. Another major weakness is that the paper structure does not properly highlight the new parts of this work. In my opinion, the implementation details of how the features are extracted, the classifier architectures, the training hyperparameters, etc., can be significantly shortened and partially put into the appendix. On the other hand, the paper can focus more on the safeguard scheme design, which includes more new insights compared to other parts.

3. Besides the experimental results, I would like to see more discussion and analysis about each set of OOD detection experiments. For example, any intuition or analysis on why features coming from certain layers are better than others with Gaussian Kernel Density and Normalizing Flows? When does Normalizing Flows consistently outperform other baselines? Why are some attack types easier to detect in the OOD setting, and what are some common generalizable features?

---

> ### Author Response · Authors · 2025-01-22
> **Response to Reviewer**
>
> We thank the reviewer for their thoughtful feedback and valuable insights on our work. We are encouraged by the recognition that OOD detection in the context of network intrusion is an important and under-explored area. As the reviewer noted, our extensive evaluation across attack types, OOD detection techniques, and neural network architectures provides significant value in demonstrating the complex interactions between these factors.
>
> ## **Weaknesses and Suggested Changes:**
> ### **Novelty in Machine Learning Techniques:**
> 1) While the neural network architectures and OOD detection techniques used in our work are based on existing methods, the novelty of our contribution lies in their application to the challenging and practical problem of network intrusion detection. This new application and associated modality (packet-level data) pose new challenges, and our work offers valuable insights on how to build a safeguard for ML-based network intrusion detection systems.  Our work highlights unique challenges of this domain, such as the dramatic performance drop in zero-day attack scenarios and the varied effectiveness of OOD techniques like Normalizing Flows across different attack types.
> 2) The release of the new packet-level dataset for network intrusion detection will be a useful resource for the research community. This is one of the primary contributions of the paper.
> ### **Paper Structure and Highlighting Novelty:**
> We appreciate the suggestion to streamline the implementation details and refocus the narrative to better emphasize our work's novelty. In the revised manuscript, we expanded the discussion of our safeguard scheme design and its implications for improving OOD detection in network intrusion settings, as this represents one of the key novel contributions of our work.
> ### **Discussion and Analysis of OOD Detection Experiments:**
> We acknowledge the need for a deeper analysis of the experimental results. In the revised paper, we include:
> 1) Intuition and analysis on why certain features (e.g., those from specific layers) perform better for Gaussian Kernel Density and Normalizing Flows, particularly in relation to feature collapse (Section 6.4).
> 2) Detailed examination of scenarios where Normalizing Flows outperform other baselines, including the characteristics of attack types that influence performance (Section 6.4).
> 3) Insights into why some attack types are more challenging to detect in the OOD setting and the generalizable features that contribute to their detectability. Brief discussion on payload length and the success of the FNN and CNN + safeguard models (Section 6.2/Section 6.4).
>
> We again thank the reviewer for their constructive comments and hope our revisions will adequately address the points raised.

---

### Author Response · Authors · 2025-01-22
**Thank You Reviewers**

We thank the reviewers for their thoughtful feedback. We have used the rebuttal period to:

1) Expand our baseline OOD models to include energy-based detectors as a potential model safeguard.
2) Provide a short discussion in the Appendix (with results) on using ODIN and the difficulties with it in settings with zero day exploits.
3) Conduct ablation studies on the normalizing flows architecture.
4) Expand discussion and analysis of the model safeguards.
5) Update the text throughout to better highlight our contributions.
6) Further elaborate on the difficulties of creating packet-level datasets from flow-level ones (with code snippets).

We will reply to each reviewer individually with more detailed comments.

---

### Decision · Action_Editor_gwwr · 2025-02-18

**Recommendation:** Accept as is

**Comment:**

All three reviewers recommend "leaning accept".   Generally, they think the application of OOD to detecting novel network attacks is interesting.  They agreed that novelty is not high since transformer and normalizing flows are existing techniques.  To strengthen the paper's contributions, the authors analyzed different novel attacks, performed ablation studies, and will be providing the datasets to the community.

**Audience:**

Individuals in the TMLR community who are interested in OOD application could find this paper interesting.

**Claims And Evidence:**

The authors proposed using an encoder-only transformer for network intrusion detection at the packet level, particularly in the out of distribution (OOD) setting for detecting novel attacks.   They proposed using a generative model with normalizing flows for OOD and compared it with energy-based and other methods.  Based on two datasets, their empirical results indicate that their approach can detect novel attacks more accurately.